# The GGDEF protein Dgc2 suppresses both motility and biofilm formation in the filamentous cyanobacterium *Leptolyngbya boryana*

Kazuma Toida,[1] Wakana Kushida,[1] Hiroki Yamamoto,[1] Kyoka Yamamoto,[1] Kaichi Ishii,[1] Kazuma Uesaka,[2] Robert A. Kanaly,[3] Shinsuke Kutsuna,[3] Kunio Ihara,[2] Yuichi Fujita,[4] Hideo Iwasaki[1,5]

**ABSTRACT**  Colony pattern formations of bacteria with motility manifest complicated morphological self-organization phenomena. *Leptolyngbya boryana* is a filamentous cyanobacterium, which has been used as a genetic model organism for studying metabolism including photosynthesis and nitrogen fixation. A widely used type strain [wild type (WT) in this article] of this species has not been reported to show any motile activity. However, we isolated a spontaneous mutant strain that shows active motility (gliding activity) to give rise to complicated colony patterns, including comet-like wandering clusters and disk-like rotating vortices on solid media. Whole-genome resequencing identified multiple mutations in the genome of the mutant strain. We confirmed that inactivation of the candidate gene *dgc2* (*LBDG_02920*) in the WT background was sufficient to give rise to motility and morphologically complex colony patterns. This gene encodes a protein containing the GGDEF motif which is conserved at the catalytic domain of diguanylate cyclase (DGC). Although DGC has been reported to be involved in biofilm formation, the *dgc2* mutant significantly facilitated biofilm formation, suggesting a role for the *dgc2* gene in suppressing both gliding motility and biofilm formation. Thus, *Leptolyngbya* is expected to be an excellent genetic model for studying dynamic colony pattern formation and to provide novel insights into the role of DGC family genes in biofilm formation.

**IMPORTANCE**  Self-propelled bacteria often exhibit complex collective behaviors, such as formation of dense-moving clusters, which are exemplified by wandering comet-like and rotating disk-like colonies; however, the molecular details of how these structures are formed are scant. We found that a strain of the filamentous cyanobacterium *Leptolyngbya* deficient in the GGDEF protein gene *dgc2* elicits motility and complex and dynamic colony pattern formation, including comet-like and disk-like clusters. Although c-di-GMP has been reported to activate biofilm formation in some bacterial species, disruption of *dgc2* unexpectedly enhanced it, suggesting a novel role for this GGDEF protein for inhibiting both colony pattern formation and biofilm formation.

**KEYWORDS**  cyanobacteria, motility, biofilms, colony pattern formation, c-di-GMP, guanylate cyclase, *Leptolyngbya*

Many bacterial species exhibit morphological colony pattern formation with motility. For example, pioneering analyses have shown that, depending on nutritional conditions, *Bacillus* shows a variety of forms, including circular, arborescent, and ring shaped (1, 2). In *Paenibacillus vortex*, while growing tree like, many small whorls form at the apical portion, resulting in extremely complex morphology (3–6). Colony patterns in *Myxococcus* and *Pseudomonas aeruginosa* are also well studied (7–10). These bacteria

Address correspondence to Hideo Iwasaki, hideo-iwasaki@waseda.jp.

The authors declare no conflict of interest.

See the funding table on p. 24.

provide excellent models for studying the collective behavior and emergent morphogenesis of organisms commonly seen across species. The basis for the formation of complex morphological patterns is that the behavior patterns, such as cell movement and orientation, change from time to time through their interactions with other motile cells and the environment. To decipher these changes, it is necessary to combine qualitative and quantitative observations of dynamics under the microscope, simulations aided by mathematical models, and molecular genetic analysis of molecular mechanisms in an integrated manner.

In flagella-less cyanobacteria, twitching and gliding motility have been studied in some species, such as *Oscillatoria* (11, 12), *Phormidium* (13, 14), hormogonia of *Nostoc punctiforme* (15, 16), *Synechocystis* sp. PCC 6803 (*Synechocystis*, hereafter) (17, 18), and *Synechococcus* (19). Twitching motility involves surface crawling through extension and contraction of type IV pilus (20). Gliding of filamentous cyanobacteria is suggested to be driven primarily by type IV-pili (T4P) supported by a polysaccharide secretion system known as the junctional pore complex (JPC). In hormogonia of *N. punctiforme,* JPC is formed with an arrayed ring structure of type IV- pilus-like systems, which in part consist of Pil, Hmp (hormogonium motility and polysaccharide), and Hps (hormogonium polysaccharide) proteins (15, 21–23). The arrayed ring accompanying JPC has also been observed in *Phormidium* species, which forms spiral clusters (11, 16, 24). We recently reported that the filamentous cyanobacterium *Pseudanabaena* sp. NIES-4403 (hereafter, *Pseudanabaena*) generates high-density migrating clusters, which were categorized into comet-like wandering and disk-like rotating clusters (25). The combination of these wandering and rotating clusters has been reported in *Bacillus* (2, 26), *Paenibacillus vortex* in the presence of mitomycin C (5), and *Paenibacillus* sp. NAIST15-1 (6). Our previous observations in *Pseudanabaena* suggested that the following processes are key to generate the wandering and rotating clusters: (i) follow-up motion of bacterial filaments via polysaccharide secretion or groove formation on solid-phase surface, (ii) formation of bundles with nematic alignment, (iii) formation of comet-like clusters accompanied by the creation of a cover of filaments at the tip in the direction of traveling, (iv) formation of rotating disks by spontaneous self-tracking of comet-like clusters, and (v) transition based on collisions between different types of clusters (25). Nevertheless, molecular details of generating these colony patterns remain largely unknown. Genetic studies have been limited to *Paenibacillus* sp. NAIST15-1 (6), where a large extracellular protein, CmoA, was identified to facilitate motility and migrating cluster formation while its biochemical property is not known.

*Leptolyngbya boryana* (*Leptolyngbya*, hereafter), previously known as *Plectonema boryanum*, is a non-heterocystous, filamentous cyanobacterium belonging to section III (27). This species is able to fix nitrogen under microoxic conditions (28) and grow heterotrophically in the presence of glucose in the dark (29). Since it is genetically tractable, *Leptolyngbya* provides an excellent model to study photosynthesis and nitrogen fixation (30–32). The *dg5* strain, referred to as the wild-type (WT) strain in this study, is a variant which is able to grow heterotrophically under complete darkness in the presence of glucose (31). Although this strain has not been reported to show active motility, we isolated a spontaneous mutant strain, named E22m1′, which showed active gliding motility and colony morphology, with a mixture of comet-like wandering clusters and disk-like rotating clusters on the surface of solid media. Comparison of genomic DNA sequences between WT and E22m1′ strains led to the identification of multiple mutations. We found that disruption of a gene named *dgc2*, which encodes a putative cyclic di-GMP (c-di-GMP) synthetase (diguanylate cyclase, or DGC), activates gliding motility. DGC catalyzes the synthesis of c-di-GMP, an intracellular signaling molecule, from two molecules of GTP. The catalytic site of DGC consists of a characteristic amino acid sequence motif, GGDEF or GGEEF (33), referred to as the GGDEF motif. c-di-GMP-dependent inhibition of cell motility has been reported in many bacterial species, such as *Escherichia coli*, *Pseudomonas aeruginosa*, and *Acinetobacter baumannii* (34–36). In cyanobacteria, Cph2 is known as DGC in *Synechocystis* for c-di-GMP production,

which inhibits pili-based motility (37, 38). Cph2 harbors a photoreceptive GAF domain that controls phototaxis in *Synechocystis* (38). Moreover, c-di-GMP has been reported to facilitate biofilm formation in many bacterial species (39, 40). In the thermophilic cyanobacterium *Thermosynechococcus vulcanus,* c-di-GMP directly binds to a cellulose synthase to facilitate cellulose synthesis and cell aggregation (41, 42).

## RESULTS

### Nullification of *dgc2* led to motility in *Leptolyngbya*

The wild-type strain of *Leptolyngbya* shows little or no gliding motility (Movie S1, upper left panel). When the cell suspension was inoculated at the center of solid media on a 90-mm dish, most grown cells remained at the inoculation spot. Within the colonies, WT filaments intertwined with each other, forming dense knot-like clumps locally. At the edges, some elongated filaments protruded, exhibiting an intertwined ivy-like morphological pattern (Fig. 1A, top-left). By contrast, the spontaneous mutant that we isolated, tentatively named E22m1′, was motile on the agar plate (Fig. 1A, top-right; Movie S1). E22m1′ was isolated as a spontaneous mutant from the non-motile, *dg5*-derived strain E22m1-dg5. E22m1-dg5—a transformant in which the kanamycin resistance gene was inserted between open reading frames (ORFs) *LBDG_21990 and LBDG_22000*—did not show any phenotypic changes. After about 1.5 y of continuous passaging of the E22m1-dg5 strain on BG-11 agar plates, we found that a subpopulation of cells became motile on agar plates. We established the motile spontaneous mutant strain designated as E22m1′. Filaments of E22m1′ rapidly spread over the plate and formed a complicated morphology, with a combination of dots and bundles of bacterial filaments (Fig. 1A; Movie S1, upper right panel). Whole-genome sequencing identified five mutations in the genome of E22m1′, in addition to the kanamycin resistance gene insertion (Table 1). A mutation was mapped to *LBDG_02920*, encoding a putative DGC with insertion of IS200-like transposon (Fig. 1B). Another mutation was a silent mutation mapped to *LBDG_25700*, encoding an unknown function protein without changing the amino acid sequence of the protein. The other three mutations were mapped to intergenic regions. Thus, we focused on the first mutation, namely, that on *LBDG_02920* and tested whether this gene is responsible for the motility and morphological phenotype of E22m1′. We found nine genes encoding proteins harboring the GGDEF motif (Fig. S1) in the *Leptolyngbya* genome. The nine genes were tentatively named *dgc1–9* in the order of the annotated nucleotide number on the genomic DNA, thereby designating *LBDG_02920* as *dgc2*. Note that the nomenclature of bacteria states that it is more appropriate to use alphabetic rather than numeric characters. However, in cyanobacteria and other bacteria, some different GGDEF genes have been designated *dgcA* etc., even though they are not homologous. Therefore, to avoid confusion, we have assigned numbers to the genes for convenience.

**TABLE 1** Polymorphism between WT (*dg5*) and E22m1′ (*dgc2⁻*) genomic DNA[a]

| Nucleotide loci on the genome | ORF ID | Putative function of the encoded protein | Mutation | Mutation on protein | Length of coding protein (residues) |
|---|---|---|---|---|---|
| 324676–324948 | *LBDG_02920* | Diguanylate cyclase | IS200 insertion | Leu456fs | 732 |
| 2754184 | *LBDG_25700* | Hypothetical protein | A to G substitution | Gly76Gly | 183 |
| 1535162–1535163 | | Non-coding region | 2-Base (AA) deletion | | |
| 2368755 | | | Neomycin phosphatase (kanamycin resistance gene) | | |
| 2431019–2431020 | | Non-coding region | 1-Base (T) insertion | | |
| 2630555–2630556 | | Non-coding region | 1-Base (A) insertion | | |

[a]Resequencing of the genomic DNA from E22m1′ (*dgc2⁻*) revealed polymorphism at six loci, including a frameshift mutation caused by the insertion of the IS200 transposon in *LBDG_02920* (Leu456fs) and insertion of the neomycin phosphatase (kanamycin resistance) gene at the genomic locus of 2368755. The insertion of the kanamycin resistance gene was artificially performed during the isolation of the mutant E22m1, which was the source of the motility mutant strain. Another nucleotide substitution of adenine to guanine was found in *LBDG_25700* at the 2,754,184 loci, which is a silent mutation that did not change the glycine residue. The other mutations found in intergenic regions are as follows: deletions of two bases (adenines) at 135,162 and 153,563; thymine insertion between 2,431,019 and 2,431,020; and adenine insertion between 2,630,555 and 2,630,556.

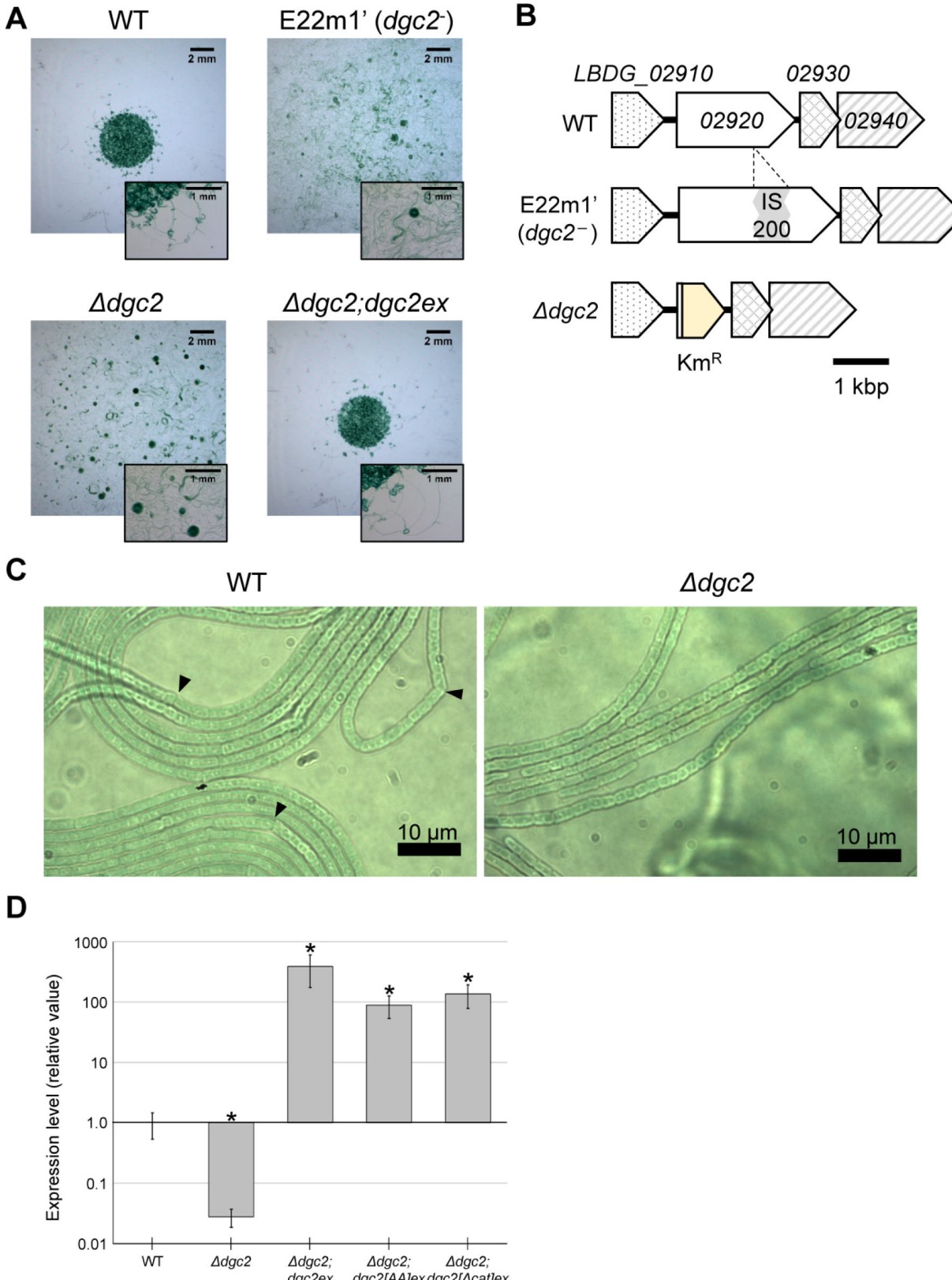

**FIG 1** Disruption of *dgc2* allowed gliding motility and dot-like colony pattern formation in *Leptolyngbya*. (A) Morphological patterns of the WT (left), *dgc2⁻* (middle left), Δ*dgc2* (middle right), and Δ*dgc2; dgc2ex* strains on solid media at 9 d after inoculation with 3 µL cell suspension at the center of culture plates. Images with magnified scale are shown on the lower right side of each panel. Both *dgc2⁻* and Δ*dgc2* strains showed gliding motility, spread rapidly on the

**FIG 1** (Continued)

solid media, and formed wandering and rotating clusters. (B) Schematic representation of genomic DNA around the *dgc2* (LBDG_02920) locus of the WT, *dgc2,* *dgc2⁻*, and *Δdgc2* strains. IS200-insertion in the ORF of *dgc2* in E22m1′ is shown in the gray box. Km^R (the yellow box) represents kanamycin resistance gene. (C) Morphological features of cells and filaments in the WT and *Δdgc2* strains on agar plates grown for 20 d. Arrowheads indicate bending points in the WT filaments. (D) Expression profiles of *dgc2* in strains by RT-qPCR analysis in the WT, *Δdgc2*, and ectopic *dgc2*-expressing strains. Asterisk represents a significant difference ($P < 0.05$ Wilcoxon rank-sum test).

To validate whether the mutation in *dgc2* is responsible for motility, we substituted the *dgc2* ORF with the kanamycin resistance gene (Km^R) in the genome of the WT (*dg5*) strain to produce the *Δdgc2* strain (Fig. 1B). The *Δdgc2* strain regained gliding motility and morphological patterns essentially similar to that of E22m1′ on solid media (Fig. 1A; Movie S1 lower left panel). No significant difference in cell morphology was observed at higher magnification (Fig. 1C). It should be noted that gliding motility in both *dgc2* mutant strains was not phototactic: when illumination was given from a specific direction, no gliding motion of the filament toward or against the light was observed. In the WT strain, very long filaments were often bundled and curved, with bends in some places (arrowheads in Fig. 1C). This is presumably due to distortion caused by cell growth. On the other hand, the lengths of bundle-forming filaments in the *Δdgc2* varied more considerably. It is possible that the intercellular space in the filaments is more fragile and prone to tearing. In addition, it also seems that the load from the various forces exerted by the movement of the cells often causes the filaments to break in the middle. Then, we expressed *dgc2* ectopically (*dgc2ex*) in the *Δdgc2* background (*Δdgc2;dgc2ex*) to test whether it complements the *Δdgc2* phenotyp. We modified a shuttle vector pPBHLI18 (43), harboring the replication origin of pGL3 in *Leptolyngbya* (44) and the *T5* promoter, which is used for overexpressing target genes (Fig. S2A). As expected, the *Δdgc2;dgc2ex* strain failed to show gliding motility, and their colony morphology bore a close resemblance to that of the WT strain—cells grew at the inoculation point and showed ivy-like pattern at the edge (Fig. 1A; Movie S1 lower right panel). RT-qPCR analysis revealed that the level of *dgc2* transcript in the *Δdgc2;dgc2ex* strain was ~700-fold higher than that of the wild-type strain (Fig. 1D). All these results strongly support that motility and colony pattern in E22m1′ are due to nullification of *dgc2*; in other words, *dgc2* represses gliding motility in *Leptolyngbya*. Thus, hereafter, the original mutant strain E22m1′ is referred to as the *dgc2⁻* strain, which is genetically different from the *Δdgc2* strain.

Then, we evaluated the movement of the wild-type, *Δdgc2*, and *Δdgc2;dgc2ex* strains at the single filament level (Fig. 2A, Fig. S2B; Movie S1). The day after inoculation on agar plates, we observed the plates under an optical microscope. We calculated the average speed of motion over a 10-min period as shown in Fig. 2B. The WT and *Δdgc2;dgc2ex* strains were almost immotile (Fig. 2A; Fig. S2B#1), whereas the *Δdgc2* strain showed gliding movement on solid media (Fig. 2A). No significant changes in filament morphology were observed in all three strains. The median velocity of motility in the *Δdgc2* strain was approximately 20 µm/min (Table 2), although one of the 28 filaments of the *Δdgc2* strain had a slow gliding speed (~0.9 µm/min). Only a few filaments of the *Δdgc2;dgc2ex* strain exhibited slight gliding (Fig. S2B#2). Multiple comparisons using the Tukey–Kramer test revealed significant differences between the WT and *Δdgc2* strains and between the *Δdgc2;dgc2ex* and *Δdgc2* strains ($P = 0.00$), whereas no significant differences were observed between the WT and *Δdgc2;dgc2ex* strains. Thus, most filaments of the *Δdgc2* strain perform gliding motility, whereas most filaments of the *Δdgc2;dgc2ex* strain do not. It should be noted that the filaments of the WT strain exhibited non-zero velocity of motion. Since the velocity shown in Fig. 2B is calculated based on the position of the center of gravity, there are two possible reasons for this: Initially, the WT strain may be able to move, although very slowly. In fact, the long filaments of the WT strain in Movie S1 appear to show slight movement. Alternatively, however, the slow movement seen specifically in long filaments can also be interpreted as passive movement due to filament distortion and pushing against each other resulting from the continuous

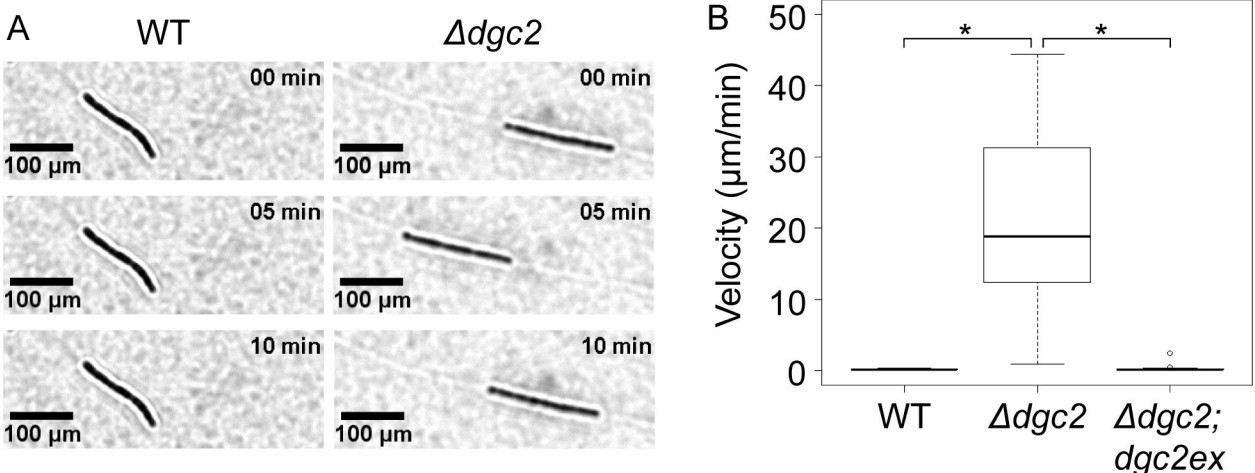

**FIG 2** Motility of the wild-type, Δ*dgc2*, and Δ*dgc2;dgc2ex* strains. A filament of the WT strain was almost immotile (A). A filament of the Δ*dgc2* strain glided and turned the direction of the movement. (B) Velocity of movement (the speed of movement of the center of gravity of the filaments) of the wild-type (*n* = 14), Δ*dgc2* (*n* = 28), and Δ*dgc2;dgc2ex* (*n* = 17) strains. There were no significant differences between WT and Δ*dgc2;dgc2*ex strains, but there were significant differences between WT and Δ*dgc2* strains and between Δ*dgc2* and Δ*dgc2;dgc2ex* strains. (*P* < 0.05 by the Tukey–Kramer test. Tables 2 and 3).

**TABLE 2** Gliding velocities of wild-type, Δ*dgc2*, and Δ*dgc2;dgc2ex* strains

|  | *WT* | Δ*dgc2* | Δ*dgc2;dgc2ex* |
|---|---|---|---|
| Max | $3.4 \times 10^{-1}$ (µm/min) | $4.4 \times 10^{1}$ (µm/min) | $2.4 \times 10^{0}$ (µm/min) |
| Median | $1.9 \times 10^{-1}$ | $1.9 \times 10^{1}$ | $1.9 \times 10^{-1}$ |
| Mean | $1.9 \times 10^{-1}$ | $2.1 \times 10^{1}$ | $3.3 \times 10^{-1}$ |
| Min | $8.3 \times 10^{-2}$ | $9.2 \times 10^{-1}$ | $9.1 \times 10^{-2}$ |

**TABLE 3** Results of the Tukey–Kramer test on the gliding velocities of the three strains

|  | *P* value |
|---|---|
| WT—Δ*dgc2* | 0.00 |
| WT—Δ*dgc2;dgc2ex* | 1.00 |
| Δ*dgc2*—Δ*dgc2;dgc2ex* | 0.00 |

filament elongation that occurs as a result of asynchronous cell division. For example, one could interpret the development of the ivy-like pattern seen in wild strains as the result of very slow gliding movement, but it could also be explained by the outward growth of the colony as rigid filaments are gradually pushed outward in an entangled manner, even if not as active motility. To address which is more plausible, we observed the WT strain in the same field of view for 100 h immediately after inoculation (Fig. S3; Movie S2). The results for 25-h observation are shown in Fig. S3A. Most of the filaments remained in the same position at 0 h, except for a few filaments that appeared at the edge of the screen. For the final stage, the positions of the filaments were verified by superimposing them on the image taken at 0 h (Fig. S3B). We colored the panel at 0 h in magenta and the one at 75 h in cyan (note: the focus was manually adjusted several times during video recording, and the position of the filament was shifted by ~3 pixels in the vertical and horizontal directions). On the other hand, the Δ*dgc2* filaments changed their positions significantly (Fig. S3C and S3D). Our results are consistent with the alternative hypothesis that entanglement and extrusion associated with the growth of rigid filaments are involved in the movement in the WT strain, and it would be more reasonable to assume that these properties are involved in the formation of the ivy-like pattern. However, we cannot rule out the possibility at this moment that a small amount of active motility that may remain would play a secondary role in this process.

In addition, Movie S2 shows the filaments in the Δ*dgc2* strain appear to alternate between relatively active phase and low phase in motility. Even in the phase of active motility, filaments with active movement and that with very low movement are observed simultaneously. Looking more closely, during hours 0–11 after the start of the measurement, the short filaments move relatively slowly in one direction. At first glance, the movement appeared to be passive, as if they were being swept along. We cannot rule out the possibility that the filaments were passively flowing because the water droplets from the spotting continued to remain to some extent at this time. In contrast, from around hour 12, most of filaments clearly began to actively glide. Filaments were often observed to change direction. In particular, several long filaments were observed to bend or become stuck as both ends moved in different directions even within each filament. At hours 40–60, the velocity in gliding motility of most filaments became relatively lowered, while at around hour 65, many filaments became actively moving again; thereafter, no simultaneous decrease in the velocity in motility of many filaments on solid media was observed until the cells began to cover the media for more than 10 d. The lack of motility immediately after inoculation may be due to filament breakage by pipetting or other processes during the preparation of cell suspensions for spotting, or to the detachment of motility machinery (pili) on the cell surface. It does not seem surprising that it takes about 12 h for these to be repaired and motility restored. On the other hand, it is not clear why cells that once showed active motility became transiently less active again for about 20 h. The period length in the alternation of the active/inactive motility phases seems to be considerably longer than that of the circadian rhythms. If the cell division cycles are synchronized among filaments for a while after inoculation and if motility is gated by the cell division cycle, the observed phase alternation would be explained while its experimental validation remains to be followed.

Interestingly, when both strains were incubated in complete darkness for ~1 mo, the Δ*dgc2* strain formed ivy-like colonies similar to the WT strain (Fig. S4A). The wild-type strain maintained the ivy-like colony and expanded 5 d later under constant light. By contrast, in the Δ*dgc2* strain, the pattern at the outer edge collapsed and expanded to fill the virgin area (Fig. S4B and S4C). This "wild type" was isolated as a mutant capable of complete heterotrophic growth (29). Although this property was not lost in the Δ*dgc2* strain, gliding motility likely requires light stimulation.

## Role for the GGDEF domain for Dgc2 function

Dgc2 consists of the cyclase/histidine kinase-associated sensing extracellular-2 (CHASE2) domain, transmembrane motifs, the Per-Arnt-Sim (PAS) domain, the PAS-associated C-terminal (PAC) domain, low-complexity region (LCR), and the GGDEF domain in order from the N-terminus (Fig. 3A, top). CHASE2 is an extracellular module conserved among bacterial sensory proteins, with unknown ligands (45). The PAS domain is a module for protein–protein interaction or ligand binding, while the PAC motif contributes to the folding of the PAS domain (46). LCR is a region of low entropy consisting of one to several amino acid residues (47), providing flexible structures in proteins for various cellular processes such as transcription and stress responses (48, 49). As mentioned before, overexpression of *dgc2* (*dgc2ex*) suppressed the motile activity of Δ*dgc2* (Fig. 2B). To validate the significance of the GGDEF domain, we tested whether overexpression of two mutant derivatives of *dgc2* would affect the result. One derivative is an active-site mutant in which GGDEF was substituted with GGAAF (Dgc2[AA]), and the other lacks the GGDEF domain (Dgc2[Δcat]) (Fig. 3A and B). Expectedly, both mutant overexpression strains, *dgc2[AA]ex* and *dgc2[Δcat]ex*, failed to inhibit motility in the Δ*dgc2* background (Fig. 3C). However, their phenotype was not identical to that of the Δ*dgc2* strain. Compared with the Δ*dgc2* strain, the mutant overexpression strains tended to remain at the initial position of inoculation and took a longer time to grow and appear dot-like patterns: the Δ*dgc2* strain formed the dot-like pattern in 2 wk, while the mutant overexpressor strains took 3–4 wk. In particular, dot-like patterns appeared in a limited area in the Δ*dgc2;dgc2[AA]ex* strain and filaments spread out in an indiscernible pattern. Thus, the

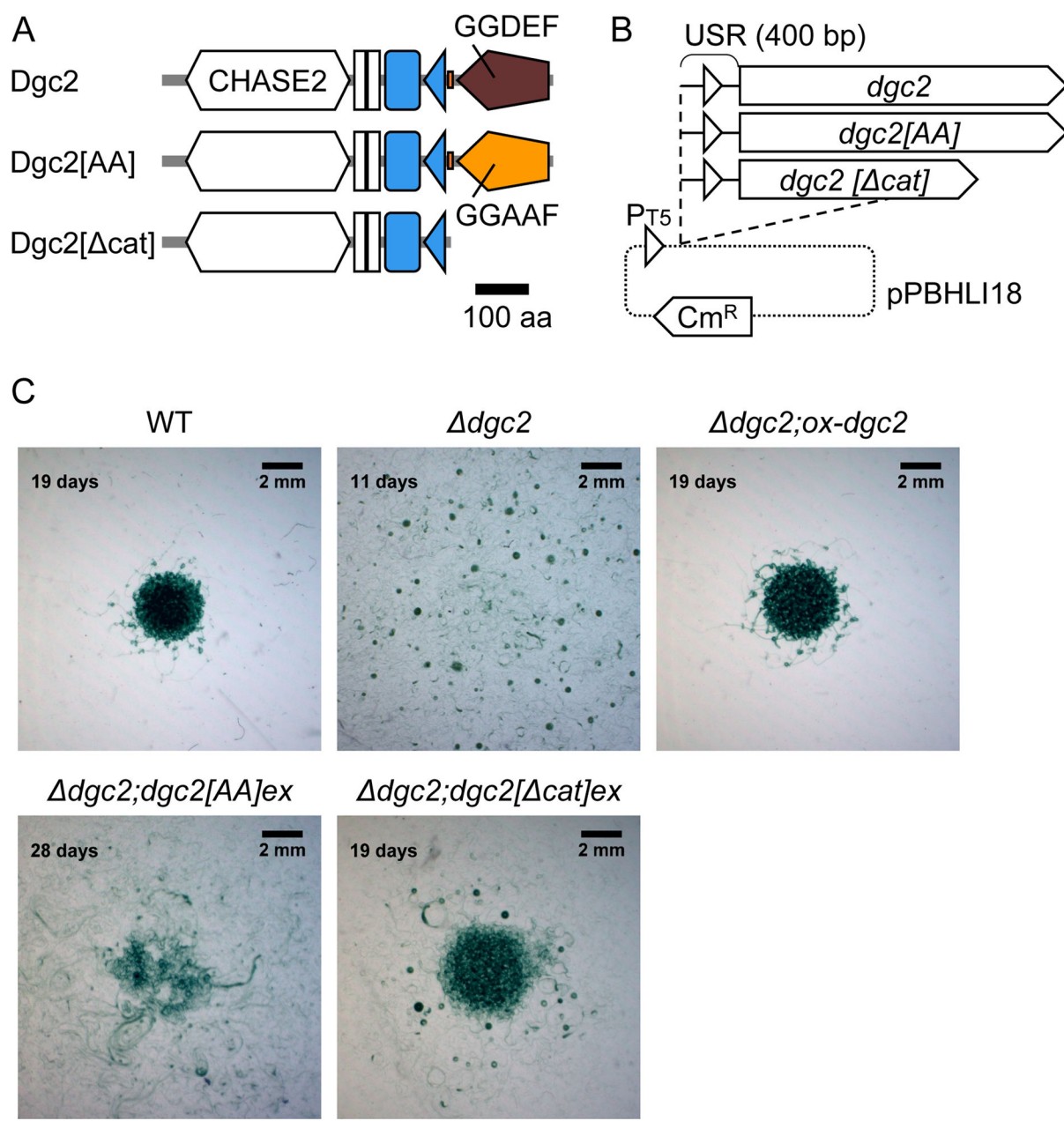

**FIG 3** GGDEF domain of Dgc2 is important for inhibiting motility and pattern formation. (A) Schematic representation of the WT and mutant Dgc2 variants for ectopic expression. CHASE2 domain (white hexagon), transmembrane motifs (white squares), PAS domain (blue square), PAC domain (blue triangle), LCR (small orange rectangle), and GGDEF motif (brown pentagon) are shown. (B) Schematic representation of plasmids for overexpressing *dgc2[AA]* and *dgc2[Δcat]*. (C) Morphological colony patterns in the mutant overexpression strains. *Δdgc2* strain is shown on day 11 from inoculation; WT, *Δdgc2;ox-dgc2*, and *Δdgc2;dgc2[Δcat]ex* strains on day 19; and *Δdgc2;dgc2[AA]ex* strain on day 28. Wild-type and *Δdgc2;ox-dgc2* strains formed an ivy-like pattern, and the *Δdgc2* strain formed a dot-like pattern. Both *Δdgc2;dgc2[Δcat]ex* and *Δdgc2;dgc2[AA]ex* showed dot-like patterns in some parts, but the centers of the colonies formed aggregates like the WT strain or were indistinct morphologically.

phenotypes of these mutant overexpression strains are likely an intermediate between the WT and *Δdgc2* strains.

The reasons for the delayed growth and limited formation of dot-like patterns in the *Δdgc2;dgc2[AA]ex* and *Δdgc2;dgc2[Δcat]ex* strains still remain obscure. Considering large excess of the transcription levels (Fig. 1D), the over-induced mutated Dgc2 would cause some dominant-negative effects. While in Dgc2[AA], the GGDEF-containing catalytic A-site (33) is mutated in the Dgc2[AA], the c-di-GMP-binding I-site (50–52) for allosteric

regulation, and N-terminal regulatory motifs, PAS and PAC or CHASE2 still remain intact. While Dgc2[Δcat] lacks both A-site and I-site, it also harbored the N-terminal regulatory domains. Thus, we cannot rule out the possibility that overexpression of these motifs would cause unexpected interactions with other nucleic acid species or proteins, thereby affecting growth and colony pattern. In any cases, the results indicate that the GGDEF domain with the catalytic A-site in Dgc2 is required to fully suppress gliding motility in *Leptolyngbya*.

Proteins with the GGDEF domain are known to catalyze a reaction that synthesize c-di-GMP from GTP. However, it is also reported that GGDEF domain proteins hydrolyze GTP to GMP (53). Therefore, we performed liquid chromatography with tandem mass spectrometry (LC-MS/MS) analysis and compared the c-di-GMP content in the WT, *Δdgc2*, and *Δdgc2;dgc2-ex* strains to test whether Dgc2 is involved in c-di-GMP synthesis. As shown in Fig. 4A, peak signals corresponding to c-di-GMP appeared at around the retention time of 15 min. When the average signal value in the WT strain is normalized to 1.0, the average signals in the *Δdgc2* and *Δdgc2;dgc2-ex* strains were 0.64 and 31, respectively (Fig. 4B). This result indicates that c-di-GMP content in the *dgc2* overexpressor strain increased about 30- to 50-fold compared to the WT or *Δdgc2* strains, suggesting that Dgc2 is actually able to catalyze c-di-GMP synthesis, and its overexpression suppressed motility through increasing c-di-GMP. Nevertheless, the degree of reduction in c-di-GMP in the *dgc2*-deficient strain relative to the wild strain is far less: while the amount of c-di-GMP in the mutant strain tended to be slightly lower than in the wild strain, but no significant difference was observed. If the reduced c-di-GMP production activity of Dgc2 is responsible for the induction of gliding movements in the *Δdgc2* strain, a somewhat more complex working hypothesis is required. For example, it is possible that disruption of *dgc2* causes a decrease in c-di-GMP concentration only in certain parts inside the cell but that the change in total intracellular c-di-GMP concentration is less severe because it is compensated by the other eight GGDEF proteins (Fig. S1). If so, would a decrease in the net amount of c-di-GMP in the WT strain lead to gliding motility? To test this possibility, we tried overexpression of a phosphodiesterase (PDE) gene, either well-studied *yhjH* from *E. coli* or its ortholog *pde1* gene from *Leptolyngbya*,

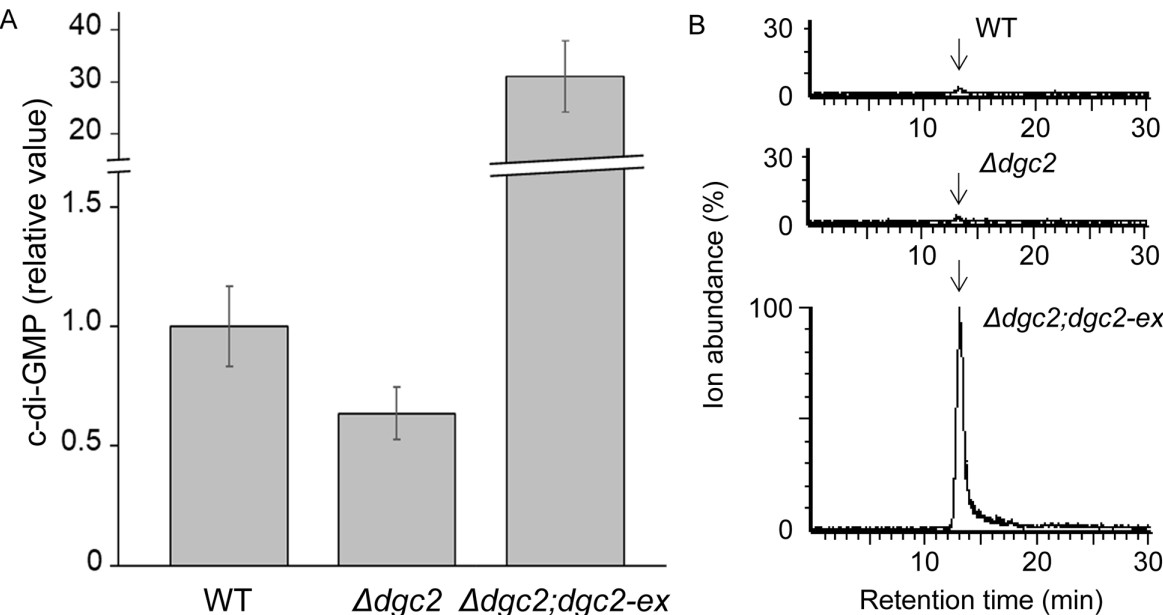

**FIG 4** Comparison of c-di-GMP levels among WT, *Δdgc2*, and *Δdgc2;dgc2ex* strains. (A) Results of LC-MS/MS analysis of c-di-GMP displaying the multiple reaction monitoring (MRM) transitions *m/z* 691 > 152. The maximum value of the signal (ion abundance) detected in the *Δdgc2;dgc2ex* strain was normalized to 100. Arrows denote c-di-GMP peaks. (B) c-di-GMP in the WT, *Δdgc2*, and *Δdgc2;dgc2ex* strains (*n* = 4), The relative levels are shown with the mean value for signals in WT strain normalized to 1.0.

in the WT strain. Both strains failed to show gliding motility in the presence or absence of isopropyl-β-D-thiogalactopyranoside (IPTG) (Fig. S5), contradicting with the hypothesis. Thus, although Dgc2 is expected to have c-di-GMP synthesis activity, no definite conclusion could be drawn as to whether the gliding activity in the *dgc2* mutant strain was due to a reduction in c-di-GMP synthesis activity.

## Effect of *dgc2* on collective behavior

The collective behavior of the *Δdgc2* strain is summarized in Fig. 5. After 9 to 12 days after inoculation, bacterial filaments spread over tens of millimeters and showed a colony pattern with a characteristic cell distribution that was spatially biased with respect to cell density (Fig. 1A, 4, and 5A; Fig. S6 and S8; Movie S3). The migrating high-density clusters were categorized into two groups: first, a comet-like wandering cluster (comet, hereafter), which migrated over solid media (Fig. S6A; Movie S4, left panel) and second, a disk-like rotating cluster (Fig. S6B; Movie S4, right panel; hereafter disk). The disk kept moving to follow closed circular orbit. A kymograph was used to indicate the time-dependent profile of collective behavior and its transition (Fig. 5B). Vertical gradient lines on the kymograph (dark red and magenta arrowheads) represent disks that remain at the position. Many disks rotated steadily over 300 min, indicated by periodically fluctuating profiles on the kymograph, consistent with those observed for *Pseudanabaena* sp. NIES-4403 (25). By contrast, disk-like clusters in *Paenibacillus* sp. NAIST15-1 immediately stopped rotating after forming large vortices (6). The original *dgc2⁻* strain also formed disks and comets (Fig. S7; Movie S3A). These results suggest that *dgc2* in *Leptolyngbya* is involved in the suppression of collective motility, thereby inhibiting the formation of migrating clusters. A transiently emerging pattern of dark short bars with a slope (blue and cyan arrowhead) on the kymograph (Fig. 5B; Fig. S7B) indicated the passage of wandering comets. Although comets of *Pseudanabaena* were proposed to be pioneers traveling to virgin areas on the solid surface (25), in *Leptolyngbya*, the comet-like cluster moved exclusively on already existing passages.

We observed some transition of clusters with collision. Fig. S6C and Movie S5A show unification of multiple comets. In this case, a preceding small comet initially formed

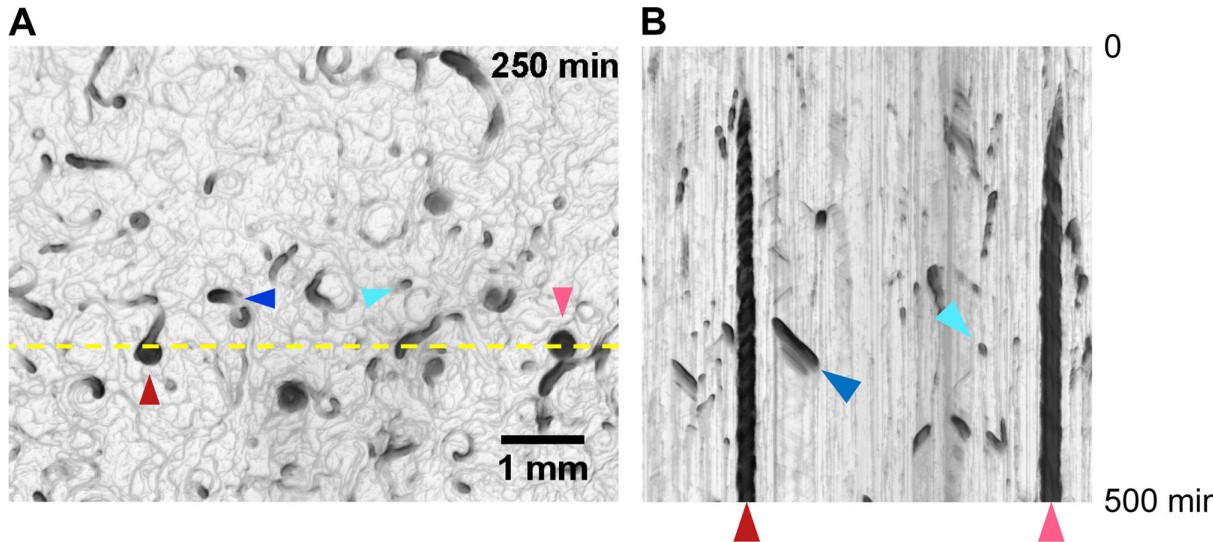

**FIG 5** Transition of collective behavior of the *Δdgc2* strain with time. Time-lapse images of *Δdgc2* strain on solid media. The analysis was performed every minute, and the images were compiled into Movie S3B. A kymograph of colonies represented by yellow dashed lines shown in panel A over 500 min (top, min 0; bottom, min 500). Blue and cyan arrowheads represent comet-like wandering clusters, whereas red and magenta arrowheads represent disks. The comet represented by a blue triangle appeared at the bottom of the field at 60 min and glided near the yellow line from 300 to 360 min, moving to the upper right side. The comet indicated by the cyan arrowhead appeared at ~4 h and passed through the line to the lower right on the field. The disks appeared at ~2 h and continued to rotate on the spot until the end of imaging. In panel B, at 500 min, comets and two disks are visible.

a circular orbit. Then, a large comet entered from behind and crossed the orbit. The small comet kept moving and then caught up with the slowly moving large comet to be unified. In the case of collision of a comet and a disk (Fig. S6D; Movie S5B), a comet moved to the direction of the tangent line of the disk's rotation and was smoothly unified into the rotating disk. We also observed a disk-to-comet transition. A group of comets slipped out from inside the disk and moved straight, where they formed a new small closed orbit within 1 h (Fig. S6E; Movie S5C).

## Counterclockwise movement is more stable for high-density clusters

Next, we analyzed the behavior of individual clusters (Fig. 5; Movie S3). For the disks that could be observed at the two angles of view (Movie S3B to D), we observed the direction of rotation for three frames from each time point and counted whether the rotation was biased in the clockwise (CW) or counterclockwise (CCW) direction (Fig. S8). Most disks always rotated in the CCW direction. Then, we made specific observations of the disks rotating in different directions and evaluated the consequences of change in direction of rotation.

Initially, we focused on disks rotating in the CCW direction, which is common in the observation (Movie S6A; Fig. 6A). The disk continued to rotate stably for 300 min, although the comets collided with it several times. Simultaneously, a kymograph was drawn to represent this situation in a static image. Because the density of cells inside the disk was not uniform, drawing the kymograph at the appropriate points indicated the direction of rotation as a gradient of shading: in case of Fig. 6A, the kymograph in the yellow box (upper side) was drawn in the lower left corner, and the kymograph in the blue box (lower side) shows a slope toward the lower right corner. Second, we focused on disks rotating in the CW direction (Movie S6B; Fig. S9A). Soon after the observation, the disk collapsed, releasing comet-like clusters. The remaining part

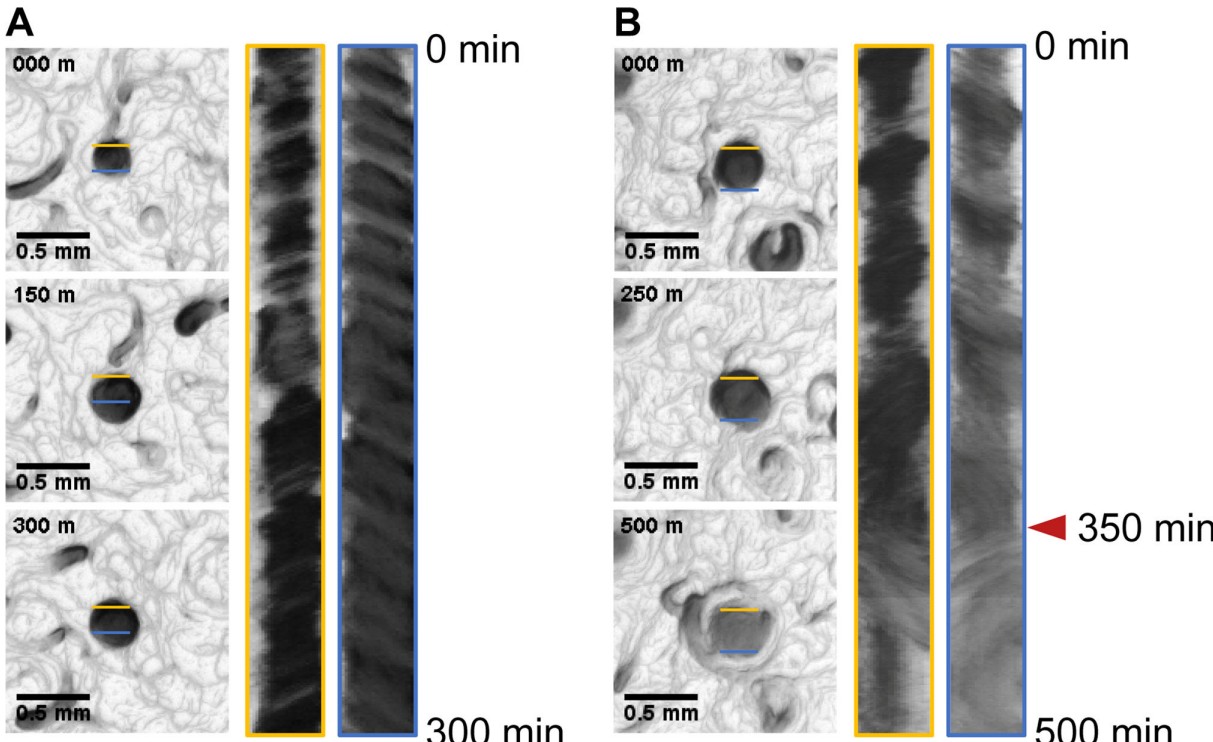

**FIG 6** Counterclockwise preference of the stably rotating disk. (A) A kymograph representing stable rotation of a disk for more than 300 min. The disk collided with three comets at ~0, 150, and 300 min. The left (yellow) and right (blue) kymographs represent time-course images at the upper and lower sides of the disk, respectively. (B) The disk initially rotated in the CCW direction, but at ~350 min (red arrowhead), the disk changed its direction of rotation to CW and eventually collapsed.

in the center exhibited almost no movement (Fig. S9A) and was considered to have stopped. Some disks changed direction during observation (Movie S6C; Fig. 6B). These disks continued to rotate in the CCW direction until ~300 min after observation began; however, the direction of rotation gradually became ambiguous and changed to CW at ~350 min (Fig. 6B, red arrowheads). The disk immediately collapsed, releasing a comet-like population. The disk suffered the same fate as the disk mentioned in the previous paragraph. Finally, we observed a disk that had continued to rotate in the CCW direction stopped rotation (Fig. S9B; Movie S7). This is the only example of a disk that was captured from its formation to its collapse. This disk had rotated CCW, but after 840 min (green arrowhead), most of the cells from the edge were separated from the disk in the form of comet. The remaining center continued rotating for a while but stopped after ~30 min. In more detail, at the initial phase (0–90 min in Fig. S9B; Movie S7), many filaments passed through a hollow circular orbit. Then, the diameter of the hollow suddenly shrank at 90–120 min to form a rotating disk. Other comets collided with the disk at ~185 min (red arrowhead) and 400 min (blue arrowhead), and a comet popped out at ~300 min (yellow arrowhead). Possibly, the impact of the collision at ~400 min was so significant that the shape of the disk changed to a distorted oval. After 840 min (green arrowhead), the disk ejected a comet and itself split into two parts. This result seems to suggest that even if the rotation is CCW, when a perturbation is too large and the angular velocity of the disk is extremely unbalanced, the disk may collapse.

Considering the CCW preference of disk-like clusters, we analyzed whether comet-like clusters also show a CCW or CW bias in the direction of motion. For time-lapse images of comets, the angle ($\theta_t$) of the velocity vector was obtained from the direction of the $X$-axis, and the angle difference ($\Phi_t$) between the angle ($\theta_t$) and next angle ($\theta_{t+1}$) was calculated. Values of $\Phi_t$ were used as indices of the perturbation angle (Fig. S10A). By calculating the duration for which the $\Phi_t$ values of taking the same sign (+ or −), we estimated how long the directionality of motion was maintained. For example, Fig. S10B shows the trajectory of migration of a comet within 663 min, and Fig. 7 represents the distribution of the duration of moving in the same direction and the average $\Phi_t$ values during that time obtained from 19 independent comets. Apparently, long-lasting (8–12 min) directional movement was limited to CCW, whereas CW motion lasted for a maximum of 7 min. Although the distribution of probability density for median

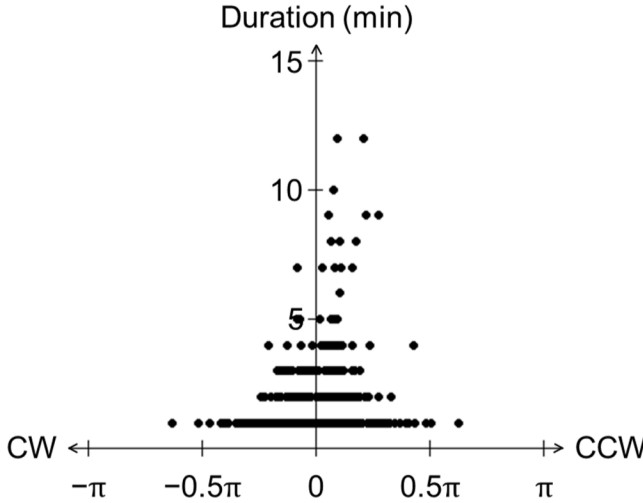

**FIG 7** Counterclockwise preference of the comet. The vertical axis indicates the duration (min) that the comet continued its clockwise or counterclockwise motion, and the horizontal axis indicates the maximum value of the progression angle at each duration. There was no significant difference in the maximum value of the progression angle between CW and CCW (Mardia–Watson–Wheeler test, $P = 0.6369$), but movement in the CCW direction was significantly longer lasting than that in the CW direction (Student's $t$-test, $P < 0.01$).

perturbation angle and duration showed no significant difference in the distribution of median perturbation angle between CCW and CW (Mardia–Watson–Wheeler test, $P = 0.6369$), the duration of the CCW direction was statistically longer than that of the CW direction (Student's $t$-test, $P < 0.01$). Interestingly, *Flavobacterium johnsoniae* was recently shown to perform gliding motility and vortex formation with CCW preference (54). It shows a CCW trajectory even from the single-cell state and remains unified in the CCW direction while rotating as a group, but moves in the CW direction just before stopping rotation. In addition, asymmetric movement for asymmetric colony pattern formation has been found in *Paenibacillus*, which has swarming motility rather than gliding motility (55). The mechanism of preference for CCW in *Leptolyngbya* remains to be studied. Note that all of these experiments were conducted in Tokyo in the Northern hemisphere. However, it is unlikely that the geomagnetic field affects the direction of rotation since the rotation trend does not change even if the agar medium is inverted from front to back.

## Extracellular polysaccharide secretion

It is possible that each bacterial cell secretes extracellular polysaccharides (EPS) during collective motility. Cyanobacteria have been reported to use T4P for gliding motility which is facilitated by the presence of EPS (11, 12, 15, 24, 25). Attempts have been made to visualize EPS. In *Oscillatoria*, *Pseudanabaena*, and *N. punctiforme*, the trajectory of the filaments was visualized using India ink, and a mucus-like substance was found on the trails (15, 25, 56, 57). Other methods, such as lectin-based ones, have been used with some cyanobacteria (15, 16, 25, 58). In this study, we stained and visualized EPS in three ways.

First, we used India ink to visualize EPS (15). We cultured the *Δdgc2* strain on BG-11 solid media for several days and allowed the cells to glide on the surface. Then, we stained the cells with India ink and observed them under a microscope for 12 min (see Materials and Methods). The trails were stained with India ink particles, confirming that the gliding filaments secreted viscous materials (Fig. 8A; Movie S8). Next, we tested whether the *Δdgc2* strain population stained with fluorescein-conjugated RCA-120, which binds to galactose and N-acetyl galactosamine residues. Positive RCA-120 signals were visualized around the filaments, confirming EPS secretion by *Δdgc2* strain (Fig. 8B). By contrast, positive signals were hardly observed in the non-motile WT strain (Fig. 8B). These results strongly suggest that the gliding activity in *Leptolyngbya* is at least partially facilitated by EPS-based mucus, and Dgc2-derived c-di-GMP production prevents EPS secretion. We also stained EPS using Ulex europaeus agglutinin I (UEA-I) from *Ulex europaeus*, which binds to glycoproteins containing α-linked fucose residues. This indicator has been previously used to stain the surface of *N. punctiforme* hormogonia as well as RCA-120 (16, 57). Although not as clear as with RCA-120, UEA-I fluorescence signals were observed at the aggregation sites of the *Δdgc2* cells (Fig. S11). Since the UEA-I was also weakly observed in the WT strain, it would also secrete a smaller number of EPS that cannot be detected by RCA-120.

## Transcriptional upregulation of genes involved in gliding motility and EPS secretion in the **Δdgc2** strain

As shown above, *Leptolyngbya* secretes EPS during gliding motility. Some genes related to T4P and hormogonium differentiation in *N. punctiforme* are also conserved in *Leptolyngbya* (21, 55, 59, 60; Table 4). To confirm whether the nullification of the *dgc2* gene actually affected the factors involved in motility, we focused on transcription of genes that are more closely related to the motility machinery and conserved in many cyanobacteria. We tested the transcript levels of *pilA*, *hmpF*, and *hpsE*, which have been reported to be involved in motility in other cyanobacteria, such as *N. punctiforme* and *Synechocystis*. PilA, called major pilin, is the main component of the pilus (20, 61, 62). Two types of ATPases, PilB and PilT, polymerize and depolymerize PilA to elongate or contract the pilus (63, 64). HmpF is an essential factor for both pilus formation and secretion (23).

## A  *Δdgc2*, India Ink

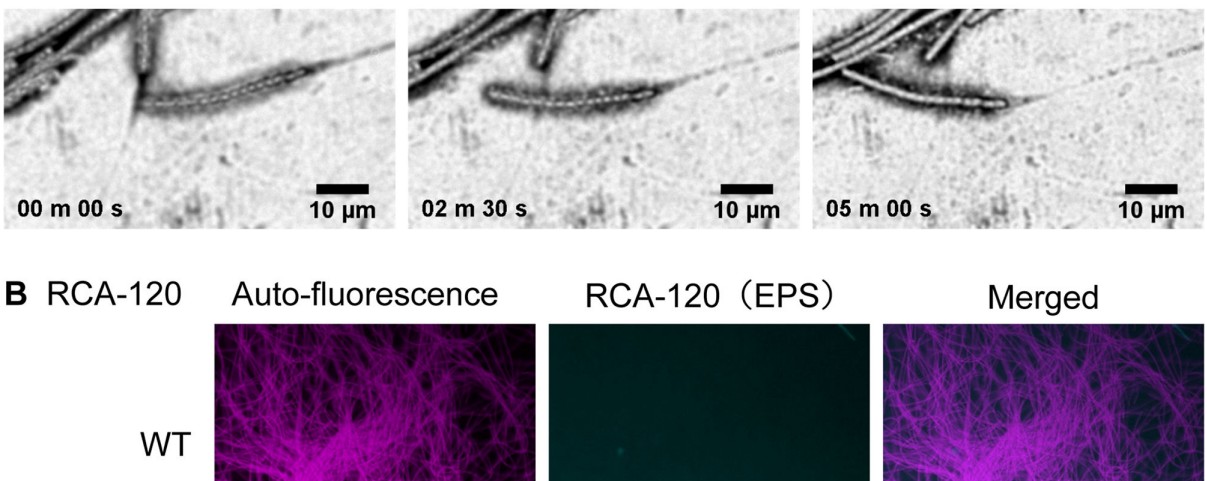

## B  RCA-120

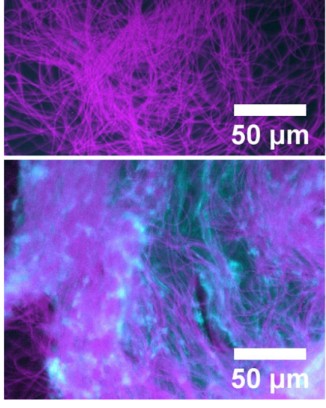

**FIG 8** The *Δdgc2* strain secretes EPS. (A) India ink staining of viscous trails of the *Δdgc2* filaments. The video is provided as Movie S8. (B) Fluorescence microscopy images of RCA-120-stained *Δdgc2* cells. Magenta and cyan indicate autofluorescence of cells and stained EPS, respectively.

It is considered to play an important role in the activation of T4P at one of the two poles of the cell. It has been reported that *hmpF* is conserved in almost all cyanobacteria except picocyanobacterial. The HmpF protein forms a fibrous structure within the cell and is involved in cell shape maintenance and motility (65). HpsE encodes a subunit of a glycosyl transferase involved in polysaccharide secretion and is widely distributed in filamentous cyanobacteria such as *N. punctiforme* (15). The *hpsEFG* operon-disrupted strain in *N. punctiforme* lost their motility. Since we wanted to analyze expression profiles in cells which show morphological patterns, RNAs were collected from cells grown on agar plates (see Materials and Methods). *dgc2* gene expression was also examined. In addition, *Leptolyngbya* does not show gliding motility under constant dark conditions (Fig. S4), indicating motility is strongly affected under diurnal cycles. Therefore, we performed RT-qPCR experiments using cells collected at dawn (after 12-h darkness) as "dark" samples, and dusk (after 12-h illumination) as "light" samples, after two 12 h:12 h light/dark (LD) cycles following to 7 d continuous illumination. It should be noted that the colony patterns on the plates were essentially similar to those reported above.

As shown in Fig. S12 and Table 5, we initially confirmed that *dgc2* expression is negligible in the *dgc2⁻* and the *Δdgc2* strain under solid media in the light. The WT strain tended to show lower levels of *dgc2* expression in the dark. While the *dgc2⁻* strain showed a faint *dgc2* transcript at even lower levels in the dark, it should be noted that the qPCR probe we used here was designed to recognize the sequence after the transposon insertion, so it does not represent a functionally intact *dgc2* transcript. Except for the *dgc2* signal in the *dgc2⁻* strain, any of the four gene transcripts show similar or lower accumulation in the dark. More importantly, the expression levels of *pilA* and *hmpF* genes in the light were significantly upregulated in both *Δdgc2* and *dgc2⁻*

**TABLE 4** Gliding-associated genes identified in *Nostoc punctiforme* have orthologs in *Leptolyngbya*[a]

| N. punctiforme | | Leptolyngbya | Identity between encoded proteins (%) | E value between encoded proteins |
|---|---|---|---|---|
| T4P | | | | |
| pilA | Npun_F0676 | LBDG_13120 | 57.4 | $1e^{-36}$ |
| pilB | Npun_R0118 | LBDG_24940 | 73.7 | 0.0 |
| pilC | Npun_R0116 | LBDG_24960 | 73.0 | 0.0 |
| pilD | Npun_R2800 | LBDG_52530 | 53.8 | $1e^{-103}$ |
| pilM | Npun_F5005 | LBDG_19450 | 67.7 | $8e^{-170}$ |
| pilN | Npun_F5006 | LBDG_19460 | 44.1 | $6e^{-61}$ |
| pilO | Npun_F5007 | LBDG_19470 | 38.6 | $3e^{-33}$ |
| pilQ | Npun_F5008 | LBDG_19480 | 46.4 | 0.0 |
| pilT1 | Npun_R0117 | LBDG_24950 | 89.6 | 0.0 |
| pilT2 | Npun_F2507 | LBDG_50840 | 69.7 | 0.0 |
| Hps | | | | |
| hpsA | Npun_F0066 | LBDG_47480 | 24.7 | $7e^{-28}$ |
| hpsB | Npun_F0067 | LBDG_47490 | 28.5 | $2e^{-11}$ |
| hpsC | Npun_F0068 | LBDG_47500 | 26.2 | $2e^{-21}$ |
| hpsD | Npun_F0069 | – | | |
| hpsE | Npun_F0070 | LBDG_29990 | 47.9 | $1e^{-109}$ |
| hpsF | Npun_F0071 | – | | |
| hpsG | Npun_F0072 | LBDG_04570 | 35.6 | $8e^{-48}$ |
| hpsH | Npun_F0073 | LBDG_47510 | 37.2 | $2e^{-28}$ |
| hpsI | Npun_F0075 | LBDG_33950 | 52.1 | $3e^{-67}$ |
| hpsJ | Npun_F0077 | LBDG_32330 | 42.0 | $6e^{-52}$ |
| hpsK | Npun_F0078 | LBDG_32320 | 74.2 | $2e^{-160}$ |
| hpsL | Npun_R0640 | LBDG_04020 | 55.4 | 0.0 |
| hpsM | Npun_R0639 | – | | |
| hpsN | Npun_R0638 | LBDG_09280 | 75.3 | 0.0 |
| hpsO | Npun_R0637 | LBDG_09290 | 70.5 | 0.0 |
| hpsP | Npun_R0636 | LBDG_09320 | 64.2 | 0.0 |
| hpsQ | Npun_F1388 | LBDG_40840 | 59.6 | $4e^{-172}$ |
| hpsR | Npun_R1506 | LBDG_18480 | 61.5 | $2e^{-129}$ |
| hpsS | Npun_R5614 | LBDG_48960 | 70.4 | $8e^{-163}$ |
| hpsT | Npun_R5613 | LBDG_55690 | 80.4 | 0.0 |
| hpsU | Npun_R5612 | LBDG_55700 | 74.9 | $4e^{-102}$ |
| Hmp | | | | |
| hmpA | Npun_F5960 | LBDG_05270 | 56.1 | $2e^{-153}$ |
| hmpB | Npun_F5961 | LBDG_05280 | 86.8 | $7e^{-75}$ |
| hmpC | Npun_F5962 | LBDG_05290 | 70.5 | $8e^{-87}$ |
| hmpD | Npun_F5963 | LBDG_05300 | 52.7 | 0.0 |
| hmpE | Npun_F5964 | LBDG_05310 | 44.2 | 0.0 |
| hmpF | Npun_R5959 | LBDG_05260 | 44.5 | $2e^{-138}$ |
| hmpU | Npun_R5135 | LBDG_41040 | 63.5 | 0.0 |
| hmpV | Npun_F5169 | LBDG_42190 | 72.3 | $5e^{-66}$ |
| hmpW | Npun_R5134 | LBDG_41050 | 67.6 | $7e^{-73}$ |

[a]Search results for genes associated with gliding motility in *N. punctiforme* that are also conserved in *Leptolyngbya*. A dash (–) indicates the absence of orthologous genes in the *Leptolyngbya* genome.

strains compared with the WT strain, while the magnitude of elevation differs. *hpsE* was significantly higher in the *dgc2⁻* strain, while it was not significantly changed in the Δ*dgc2* strain. These results support that PilA, a major T4P component, plays a major role for light-activated gliding motility, and Dgc2 negates its function via at least in part transcriptional repression of *pilA*. By contrast, even in similarly motile strains, there was variation between the Δ*dgc2* and *dgc2⁻* strains in gene expression and the rate of changes. The cause of this difference is currently unknown, but it is possible that the

**TABLE 5** Results of Dunnett test on expression levels of *dgc2*, *pilA*, *hmpF,* and *hpsE*[a]

| Tested gene | Tested data | P value |
|---|---|---|
| *dgc2* | Δ*dgc2* (light) | 0.00100 |
| | *dgc2⁻* (light) | <0.001 |
| | WT (dark) | 0.01613 |
| | Δ*dgc2* (dark) | 0.00101 |
| | *dgc2⁻* (dark) | 0.00176 |
| *pilA* | Δ*dgc2* (light) | <0.001 |
| | *dgc2⁻* (light) | 0.293 |
| | WT (dark) | 0.815 |
| | Δ*dgc2* (dark) | 0.537 |
| | *dgc2⁻* (dark) | 0.607 |
| *hmpF* | Δ*dgc2* (light) | <0.001 |
| | *dgc2⁻* (light) | 0.818 |
| | WT (dark) | 1.000 |
| | Δ*dgc2* (dark) | 0.824 |
| | *dgc2⁻* (dark) | 0.978 |
| *hpsE* | Δ*dgc2* (light) | 1.000 |
| | *dgc2⁻* (light) | <0.0001 |
| | WT (dark) | 0.994 |
| | Δ*dgc2* (dark) | 0.997 |
| | *dgc2⁻* (dark) | 0.994 |

[a]The expression level of each gene in the WT strain in the light was used as the control group, and the Dunnett test was used to test whether there was a significant difference between each group. Gene names, groups tested, and the *P* values are shown in the table.

remaining N-terminal region of Dgc2 before the transposon insertion has some effect in the *dgc2⁻* strain. Another possibility that some single-nucleotide polymorphisms other than *dgc2* may have secondary effects cannot be also ruled out.

## Dgc2 interferes with biofilm formation in *Leptolyngbya*

c-di-GMP facilitates biofilm formation in many bacterial species (39). Therefore, we examined whether biofilm formation is affected by the presence of *dgc2* in *Leptolyngbya*. We cultured *Leptolyngbya* in BG-11 liquid media on flat-bottom glass plates (60 mm diameter) without aeration or shaking. We found that WT filaments formed some aggregates and tended to sink to the bottom (Fig. 9A, WT strain with "pre"). However, when liquid media were removed using a pipette, the cells tended to become non-adherent (Fig. 9A, WT with "post"). Some filaments formed aggregates at the air–liquid interface (like flocculation). The aggregates that formed inside the flask were fragile and easily broken during pipetting or decantation of the medium. By contrast, the Δ*dgc2* strain formed larger aggregates at the bottom (Fig. 9A, Δ*dgc2* with "pre"). The aggregates were firmly attached to the bottom as stable biofilm and were difficult to disrupt when the liquid was removed with a pipette (Fig. 9A, Δ*dgc2* with "post"), even after the bottom of the plate was vigorously scraped with the tip of a pipette. We next quantified the ratio of bottom-attaching biofilm mass to total cell mass (adhesion rate), employing the method used for evaluating biofilm formation in *Synechococcus* (66). A schematic procedure of the experiment is shown in Fig. S13A. Cells were cultured on glass-bottom plates for 10 d. Then, the supernatant was collected by pipetting, and chlorophyll was extracted with methanol. Chlorophyll content was represented as *Cs*. Chlorophyll extracted from cells attached tightly to the bottom (biofilm) was quantified as *Cb* (Fig. 9B). Despite a large variation in total chlorophyll content (*Cs* + *Cb*) among the plates, the general trend was that *Cb* value was higher in the Δ*dgc2* strain than in the WT strain (Fig. 9B; Fig. S13B, *n* = 29 for both strains). In Fig. 9B, black and magenta plots indicate data for the WT and Δ*dgc2* strains, respectively. Dotted lines indicate linear approximate lines, with correlation coefficient values (Spearman's rank correlation coefficient) of 0.43

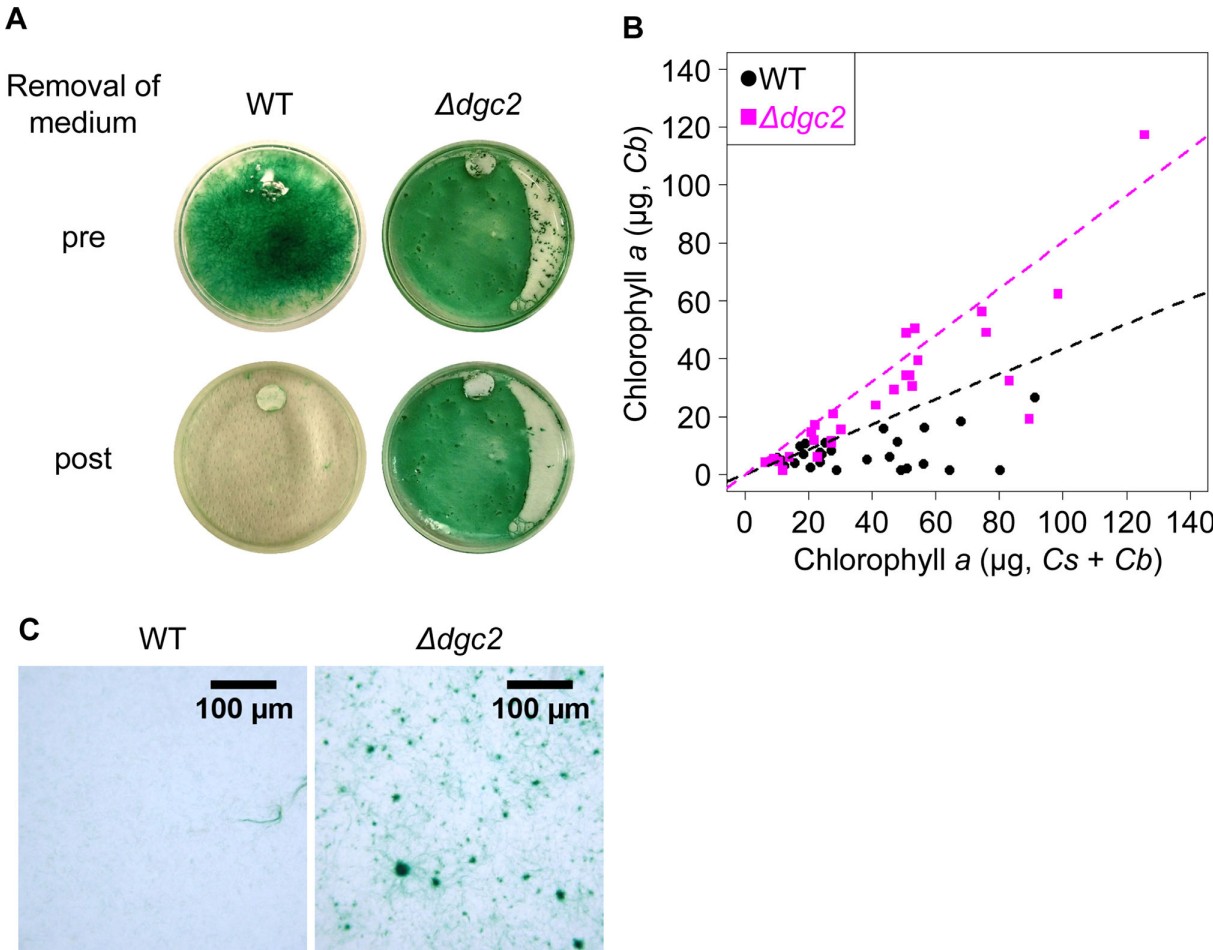

**FIG 9** Dgc2 inhibits biofilm formation in *Leptolyngbya*. (A) Representative images of aggregation/biofilm profiles in the WT and Δ*dgc2* strains before (pre) and after (post) removal of liquid media. The WT strain formed some floating aggregates that were not tightly attached to the glass surface. By contrast, the Δ*dgc2* strain developed larger aggregates that attached more tightly to the bottom (biofilm). (B) Distribution of total chlorophyll content (*Cs* + *Cb*, abscissa) and chlorophyll extracted from attached cells (*Cb*, ordinate) in each experiment. Black and magenta plots indicate data for the WT and Δ*dgc2* strains, respectively. Dotted lines indicate linear approximate lines with correlation coefficients for the Δ*dgc2* and the WT strains, respectively. (C) Microscopy images of attached cells in the WT and Δ*dgc2* strains, after the liquid phase was discarded.

and 0.80 for the WT and Δ*dgc2* strains, respectively. Notably, it seems apparent that the difference increases with increasing *Cs* value, strongly supporting a density-dependent effect in which biofilm formation on the glass bottom is promoted as cell density increases. Under the microscope, Δ*dgc2* bacterial filaments were scattered on the glass surface, forming foci (aggregates) of various sizes (Fig. 9C, right panel). The foci were formed with many filaments or bundles, extending radially outward according to their size, and connected to each other through the filaments. This morphology resembles the structure of a scale-free network of interconnected hubs of various sizes, suggesting that cell aggregation during biofilm formation involves multiplication processes characteristic of such networks. By contrast, the WT filaments rarely formed such structures; however, we observed a faint bundle or single filaments attached to the surface more randomly (Fig. 9C, left panel).

## DISCUSSION

We have identified that Dgc2 suppresses motility in *Leptolyngbya*. Although motility has not been reported in the WT strain, some *Leptolyngbya* spp. are known to show gliding motion (67). Wild isolates closely related to the immotile model organism *Synechococcus*

*elongatus* PCC 7942 were found to be motile (19), suggesting that the PCC 7942 strain lost motility during long-term culture in the laboratory, as was proposed to *Anabaena* (*Nostoc*) sp. PCC 7120 (68). Thus, the WT *Leptolyngbya* may also have been motile under natural conditions. Our results do not exclude the possibility that the WT strain may also show motility if Dgc2 function is inhibited by as yet unknown natural conditions or environmental stimuli. The presence of the CHASE2 and PAS domains in Dgc2 (Fig. 3A) would support this possibility—sensing an extracellular signal by CHASE2 and binding of a cognate partner by the PAS domain would regulate its biochemical activity. Alternatively, the WT strain may have accumulated additional spontaneous mutations to suppress motility during maintenance in laboratories over decades. As mentioned above, the WT strain used in the current study, *dg5*, is already one of the spontaneous mutant strains to grow in complete darkness in the presence of carbon sources. Although no gliding motility is observed even in this strain in the dark, it is an interesting question if gliding activity in the *dgc2*-deficient strain in the current study is unique to the *dg5* background, where metabolic levels are permanently elevated. In any case, searching for possible ligands or binding partners of Dgc2 is warranted. Notably, genes encoding Dgc2-like, CHASE2 domain-containing DGCs are found not only in cyanobacteria but also widely in the genomic DNAs of other bacterial species belonging to Proteobacteria, such as *Desulfobulbaceae*, and Verrucomicrobia, such as *Methylacidiphilum* (Table S3). Thus, the CHASE2-mediated regulation of GGDEF protein functions seems widely conserved. However, it is not entirely clear which ligands and signals are recognized by the CHASE2 domain in DGCs. The genetically tractable *Leptolyngbya* is an excellent model to address this issue.

The dramatic increase in the c-di-GMP content in *dgc2*-overexpressor strains suggests that Dgc2 is able to synthesize c-di-GMP. Nevertheless, total intracellular c-di-GMP was less affected in the *Δdgc2* strain compared to the WT strain. Moreover, the over-expression of PDE did not lead to motility with *Leptolyngbya*. These results severely challenge our original working hypothesis that Dgc2 suppresses gliding activity and the motile cluster formation by reducing intracellular c-di-GMP production. There are two alternative possibilities. The first would be that the c-di-GMP-producing nature of Dgc2 is still important but may lead to localized changes in c-di-GMP concentration at specific area of the cell. For example, for *Leptolyngbya* filaments to exhibit gliding activity, c-di-GMP level must be reduced only in a specific region, where the machineries for motility are abundant, while a considerable c-di-GMP level by other DGCs needs to be preserved elsewhere. Although it is difficult to verify this directly at present, evaluation of the subcellular localization of Dgc2 fused with a fluorescent protein would address its possibility: although we tried this experiment, we have not completed in time to verify whether the *Δdgc2* strain can be rescued by the translational fusion construct. Since nullification of *dgc2* increased the transcription of *pilA*, which encodes the major pilin, it is possible that Dgc2 suppresses motility at least in part by suppressing the transcription of *pilA*. It has been reported that the transcription of *pilA* in *Synechocystis* is dependent on the amount of c-di-GMP, while the targets are *pilA5* and *pilA9*, which belong to minor pilin, and not major pilin (69). In *P. aeruginosa,* genes involved in flagellar and biofilm formation are regulated by FleQ, a c-di-GMP-binding transcriptional regulator (70). However, it seems difficult to alter the activity of a specific transcription factor by localized change in the level of c-di-GMP.

The second alternative possibility is that Dgc2 has a major biochemical function other than c-di-GMP production; since it has the GGDEF motif, it is not surprising that it can synthesize c-di-GMP under overexpression. On the other hand, it is now known that the GGDEF motif confers GMP synthesis as a side effect, and more recently revealed that the GGDEF protein, Bd0367, in *Bdellovibrio bacteriovorus* synthesizes not only c-di-GMP but also 3′,3′-cyclic-GMP-AMP (cGAMP) (71). Although Dgc2 does not conserved Ser residues that are important for cGMAP production in Bd0367, a possibility that Dgc2 plays a unique function by synthesis of as-yet-unknown second messengers other than c-di-GMP cannot be ruled out at this time. It is also possible that Dgc2 regulates the

activity of other enzymes via the PAS and PAC domains involved in protein–protein interactions.

We also found that *Leptolyngbya* Dgc2 suppressed biofilm formation (Fig. 9), which contrasted to some reports that c-di-GMP or DGCs facilitates biofilm formation in many bacterial species (70, 72–74). This interesting difference suggests more diverse relationship between biofilm formation and motility. For example, in *Pseudomonas*, motility may act as an inhibitor of biofilm formation by decreasing the strength of cell–cell interactions through the movement of individual cells. By contrast, in the Δ*dgc2* strain of *Leptolyngbya*, activation of T4P, which is required for motility, may also promote adhesion of cells to solid-phase surfaces. Alternatively, the acquisition of motility may rather promote migration-associated grouping and, thus, enhances the strength of interaction among filaments. If the latter were true, then formation of high-density migrating clusters on the agar-based medium could be interpreted as pattern formation brought about by the physical properties of biofilm formation with motility under the artificial conditions of the flat solid surface of agar media. The formation of foci by motile filaments for biofilm formation shown in Fig. 9C may also support this possibility. Conversely, the colony patterns on solid-phase media shown in this study could actually be formed at the aqueous-gas-phase interface in natural environments, for example, at the shore regions of lakes, where the formation of comet-like wandering colonies could lead to the movement of many bacteria from an aversive environment to a new environment. On the other hand, if the environment is somewhat comfortable, it may be possible to remain there by becoming a disk colony. In practice, however, they would coexist with other species or take different forms in the water or on the bottom. We expect that major characteristics or dynamics underlying the colony pattern formations obtained here (e.g., motility characteristics of individual filaments and interactions among filaments) would also play an important function in achieving such different morphologies under natural conditions.

## MATERIALS AND METHODS

### Strains and culture conditions

Strains used in this study are summarized in Table 6. *Leptolyngbya dg5* strain, which harbors heterotrophic capability under continuous dark conditions (31), was used as the WT strain. The original motile strain E22m1′ was isolated as a spontaneous mutant, derived from a transformant strain E22m1-dg5. The E22m1-dg5 parental strain carrying the kanamycin resistance gene (27 bp downstream of the *LBDG_21990* ORF) did not show any observable phenotypes, including neither motility nor colony morphology phenotypes. The motile E22m1 strain was renamed *dgc2⁻* after genetic analysis. For generating the Δ*dgc2* strain, we ligated a 1,500-bp upstream region (USR) of

**TABLE 6** Strains used in this study

| Strain ID | Genotype | Host | Plasmid | Source |
|---|---|---|---|---|
| *dg5* | Wild type | | | Fujita et al. 1996 |
| E22m1-dg5 | A kanamycin resistance gene insertion between *LBDG_21990* and *LBDG_22000* | *dg5* | E22m1-dg5 | This study |
| E22m1′ | IS200 transposon insertion in *dgc2* (*LBDG_02920*) | E22m1-dg5 | | This study |
| ILC 1004 | Δ*dgc2* | *dg5* | pIL910 | This study |
| ILC 1407 | Δ*dgc2;dgc2ex* | ILC 1004 | pIL1007 | This study |
| ILC 1412 | *dgc2ex* | *dg5* | pIL1007 | This study |
| ILC 1440 | Δ*dgc2;dgc2[AA]ex* | ILC 1004 | pIL1018 | This study |
| ILC 1441 | Δ*dgc2;dgc2[Δcat]ex* | ILC 1004 | pIL1019 | This study |
| ILC 1486 | *dgc2[AA]ex* | *dg5* | pIL1018 | This work |
| ILC 1487 | *dgc2[Δcat]ex* | *dg5* | pIL1019 | This study |
| ILC 1827 | P*trc::pde1* | *dg5* | pIL1195 | This study |
| *ILC 1832* | P*trc::yhjH* | *dg5* | pIL1205 | This study |

*LBDG_02920*, the kanamycin resistance gene cassette from pYFC10 (26), and a 1,500-bp downstream region (DSR) of *LBDG_02920*, in this order, and then cloned these fragments into a 1,752-bp segment derived from pBluescript II SK(+) (Stratagene) to produce pIL910. For *dgc2*-overexpression strains, we cloned a segment covering a 400-bp USR of *LBDG_02920* and the ORF into the *Sph*I-*Bam*HI site of a shuttle vector pPBHLI18 (43) to generate pIL1007. pPBHLI18 is a derivative of pPBH201 (44) and harbors the *T5* promoter and terminator derived from pQE70 (Qiagen), the replication origin of pGL3 (Promega) in *Leptolyngbya*, and a chloramphenicol resistance gene cassette. Shuttle vectors harboring mutated *dgc2* (pIL1018 and pIL1019) were generated by PCR-based targeted mutagenesis of pIL1007. For generating $P_{trc}::pde1$ strain, we replace intergenic region between *LBDG_36740* and *LBDG_36750* with kanamycin resistance gene, lacI$^q$, and $P_{trc}$ sequences. We ligated a 1,500-bp USR of *pde1*, the kanamycin resistance gene cassette from pYFC10, lacI$^q$ and $P_{trc}$ from pTrc99A (75), and a 1,500-bp DSR of *pde1*, in this order, and then cloned these fragments into 2,392 bp segments derived from pBR322 to pIL1195. For generating Ptrc::yhjH strain, we inserted them into LBDG_21990 and LBDG_22000 region. We ligated a 1,500-bp USR including LBDG_21990, the kanamycin resistance gene cassette from pYFC10, lacIq and Ptrc from pTrc99A, *yhjH* from *E. coli* JM109, and a 1,500-bp DSR including LBDG_22000, in this order, and then cloned these fragments into 2,366 bp segments derived from pIL1195. DNA fragments were generated using PCR with primers listed in Table S1. The DNA segments were joined using the NEBuilder HiFi DNA Assembly Kit (NEB) or Ligation high Ver. 2 (TOYOBO). The list of constructed plasmids is summarized in Table 7. All plasmids were checked by DNA sequencing. Transformants used were generated using the electroporation method (30, 31, 76), with some modifications. Instead of the filtration method described by Tsujimoto et al. (75), we centrifuged and washed cells for collection. Deletion of genes was confirmed by Southern blotting using the DIG DNA Labeling Mix (Sigma-Aldrich). Transformation with shuttle vectors was confirmed by PCR using purified genomic DNA as template. All strains were cultured in BG-11 solid or liquid media (77) under illumination. Under the dark conditions, we supplemented 7.5 mM glucose. We used white fluorescence lamps of 7.5 µmol $m^{-2}s^{-1}$ for most experiments, while we used hat of 10 µmol $m^{-2}s^{-1}$ for experiments shown in Fig. 9; Fig. S12. For experiments shown in Fig. S4, cells were grown in darkness for 28 d and then exposed to the light of 7.5 µmol $m^{-2}s^{-1}$ for 5 d. To compare the colonies formed by each strain (Fig. 1A, 2A, 3, 4, 6A and 7C; Fig. S2B, S3-S7, S9, S10B; Movies S1 to S7), we spotted the bacterial suspensions on the agar medium. Cells grown on agar plates were collected with a platinum loop and suspended in BG-11 liquid medium to $OD_{730} = 0.1$. Then, 2 µL cell suspension was inoculated on a fresh BG-11 agar plate. The strains harboring the shuttle vector plasmid were cultured on BG-11 agar media containing chloramphenicol, whereas other strains were cultured on BG-11 agar medium without antibiotics. At the time of inoculation, the WT and *Δdgc2* strains were cultured as controls.

## EPS staining assay

EPS staining with fluorescent lectin (RCA-120 and UEA-I) was performed as described (15, 16, 78) with slight modification as follows (Fig. 6B; Fig. S11). Briefly, 100 µL of cell suspension was centrifuged at 3,500 rpm at room temperature (RT) for 5 min, and the supernatant was removed to collect cell pellet. Then, 100 µL of fresh BG-11 medium containing RCA120-fluorescein (Vector laboratories, USA) at 20 µg/mL (final concentration) was incubated with the collected cells at RT for 30 min. The cells were centrifuged at 3,500 rpm for 5 min at RT, the supernatant was removed, and the cell pellets were suspended in 100 µL of fresh medium and then placed on a slide glass. The stained samples were observed under the IX-71 inverted microscope (Olympus, Japan) with the LUCPlanFLN (20×) objective (Olympus, Japan) and filter sets for RCA120-fluorescein (Ex 485–517 nm, Em 505–565 nm) and autofluorescence (Ex 544–592 nm, Em 570–650 nm). For ink staining, we used cells cultured on BG-11 solid media for ~1 wk. Then, 10 µL of solution containing 2% India ink (Winsor & Newton, UK), 4 µM $CaCl_2$, and 0.05% (vol/vol)

**TABLE 7** Plasmids used in this study

| Plasmid | Construction | Replication origin | Antibiotic resistance marker | Host vector |
|---|---|---|---|---|
| pIL910 | USR 1500 bp of *dgc2*::Km$^R$::DSR 1500 bp of *dgc2* | ColE1 | Ap$^R$, Km$^R$ | pBlueScript SK II (+) |
| pIL985 | Sph I-dgc2-BamH I | ColE1 | Cm$^R$ | pPBHLI18 |
| pIL1007 | *Sph* I-USR400 bp of *dgc2*::*dgc2*-*Bam*H I | ColE1 | Cm$^R$ | pIL985 |
| pIL1018 | *Sph* I-USR400 bp of *dgc2*::*dgc2[AA]*-*Bam*H I | ColE1 | Cm$^R$ | pIL1007 |
| pIL1019 | *Sph* I-USR400 bp of *dgc2*::*dgc2[Δcat]*-*Bam*H I | ColE1 | Cm$^R$ | pIL1007 |
| pIL1195 | USR1500 bp of *pde1*::Km$^R$::*lacI$^q$*::P$_{trc}$::LBDG_36750 (*pde1*)::DSR 1500 bp of *pde1* | ColE1 | Ap$^R$, Km$^R$ | pBR322 |
| pIL1205 | LBDG21990_Km$^R$::*lacI$^q$*::P$_{trc}$::*yhjH*::LBDG_22000 | ColE1 | Ap$^R$, Km$^R$ | pIL1195 |

Triton X-100 (Sigma) was then dropped on the surface of the medium. Images were taken using a microscope (IX71; Olympus, Japan) with the UPlanApo (100×) objective (Olympus, Japan) before staining and 2 min after staining (Fig. 6A; Movie S8).

## Biofilm assay

The biofilm assay was performed as described (66) with some modifications as follows. Briefly, 100 µL of liquid culture of the WT or Δ*dgc2* strain was suspended to an OD$_{730}$ value of 1.0 in 10 mL of BG-11 in a glass plate (60 mm diameter). Biofilm formation was incubated at 30°C for 10 d without aeration under low light illumination, with a photon flux density of ~7.5 µmol m$^{-2}$s$^{-1}$. After incubation, cells attached to the glass bottom and cells in the liquid phase, including floc, were separately collected. Chlorophyll was extracted from samples incubated in glass plates as follows. For samples remaining in the liquid phase, the cell suspension (supernatant) was collected in a 15-mL centrifuge tube by pipetting and centrifuged at 3,500 rpm for 5 min at RT to obtain a pellet. Then, 2 mL of methanol was added to the centrifuge tube (liquid sample) or glass plate (bottom-attached sample). The mixture was stirred well and incubated for 5 min at RT. After centrifugation at 10,000 rpm for 10 min at RT, the absorbance of each supernatant at 665 nm was measured to determine chlorophyll *a* content (Fig. 9B; Fig. S13B, 78). The chlorophyll content derived from cells in the liquid phase and from those attached to the glass bottom was designated as *Cs* and *Cb*, respectively.

## Imaging of cells

Time-lapse images for colony patterns on agar plates were taken using the SZX16 microscope (Olympus, Japan) with the SDFPLAPO 0.5XPF (0.5×) objective (Olympus, Japan) and the Moticam Pro282B CCD camera (Motic, China). Motic Images Advanced 3.2 (Motic, China) was used as the control software (Fig. 1 and 7C; Fig. S5). For higher-resolution microscopy, we used the IX71 inverted microscope (Olympus, Japan) connected to a cooled CCD camera (Pixis: 1,024, Princeton Instrument, USA, or Retiga EXi Fast 1394, Qimaging, Canada), controlled by the SLIDEBOOK 4.2 software (Intelligent Imaging Innovations), except for Fig. 1C. For observations shown in Fig. 3 and 4; Fig. S6, S7, S9, and S10B; Movies S3 to S7, we used the 1.25× objective lens (PlanApo N 1.25×, N.A. 0.04, W.D. 5.1 mm, Olympus, Japan). For observations shown in Fig. 2A; Fig. S2B and S3; Movies S1 and S2, we used a 4× objective lens (Uplan FL10 × 2, N.A. 0.30, W.D. 10 mm, Olympus, Japan). The TG-5 digital camera (Olympus, Japan) was used to capture images of cells on glass plates shown in Fig. 9A and C and cells on agar plates shown in Fig. S4. ImageJ 1.52 a (NIH, USA, 79) was used for image processing and analysis. To calculate the velocity of the filaments, the position of the center of gravity was automatically selected and tracked. The displacement was recorded with each frame (1-min interval), and the average value over 10 frames was calculated and considered as the filament's velocity. For determining perturbation angles of comets, tip of each comet was manually tracked at each time points. Statistical analysis was performed using the free software R-4.0.3 [64-bit, (80)]. For passing count image shown in Fig. S10, the comet-like cluster was tracked manually at the presumed leading position and their positions were recorded.

To display the trajectory on images, we used a macro in ImageJ so that the plotted positions from tracking images continuously changed the color from magenta to cyan. For Fig. 1C, we observed cell filaments grown on agar plates using an Olympus BX50 microscope with a 100× objective lens (UPlanApo 100×/1.35 oil) and the Moticam 2500 CMOS camera (Motic, China).

## RNA analysis

Total RNA was extracted from cells grown in either liquid or solid medium. For RT-qPCR analysis shown in Fig. 1D, the WT, ILC 1004 (Δdgc2), ILC 1407 (Δdgc2;dgc2ex), ILC 1440 (Δdgc2;dgc2[AA]ex), and ILC 1441 (Δdgc2;dgc2[Δcat]ex) strains were initially inoculated into 60 mL of liquid BG-11 media (with chloramphenicol or kanamycin, if necessary) under continuous illumination with white fluorescence lamps of 20 µmol photons $m^{-2}s^{-1}$ for ~7 d, which allowed cells to enter the logarithmic phase. Cells with a bacterial mass such that the value of the multiplication of the $OD_{730}$ value by the liquid volume (mL) was 5.0 were collected. After centrifugation and removal of supernatant, the cell pellets were resuspended in 120 mL of fresh BG-11 medium supplemented with antibiotics and further incubated under continuous illumination. Cells at the logarithmic phase with an $OD_{730}$ of 1.0 ± 0.4 were collected; the value of the multiplication of the $OD_{730}$ value by the liquid volume (mL) was 3.0. Cells were immediately washed with chilled 50 mL distilled water (DW) after centrifugation at 4°C for 10 min at 3,500 rpm. The cell pellets were resuspended in 300–500 µL of chilled water in a 2-mL screw-capped tube, added with four zirconia beads (3 mm diameter, Nikkato, Japan), and then stored at −80°C until use. For RT-qPCR analysis shown in Fig. S11, each of the WT and Δdgc2 strain was grown on 60 mL of BG-11 agar medium (in the presence of kanamycin for Δdgc2) divided into two 90-mm plates for ~9 d under a constant illumination of 7.5–10 µmol $m^{-2}s^{-1}$. After the addition of 5 mL BG-11 liquid medium, cells were collected from agar surface with a cell scraper and then diluted to $OD_{730}$ of ~4. An amount of 100 µL of the cell suspension was then inoculated onto BG-11 solid media on six 144 mm × 104 mm square plates (72 mL medium per plate) for each strain. After 7 d of incubation under a continuous illumination of 7.5–10 µmol $m^{-2}s^{-1}$, the cells were exposed to two 12 h:12 h LD cycles to synchronize the circadian clock. Then, cells were collected at hours 0 and 12 in the light (as dark and light samples shown in Fig. S11, respectively) with a cell scraper from agar surface of three plates after the addition of 10 mL of chilled sterile water per plate. The cells were pelleted by centrifugation at 4°C for 5 min at 3,500 rpm, resuspended in 300–500 µL of water in a 2-mL screw-capped tube, added with four zirconia beads (3 mm diameter, Nikkato, Japan), and then stored at −80°C until use. For RNA extraction, the acid hot phenol method (19) was used with some modifications. The stored cells were crushed with the MultiBeads Shocker (Yasui Kikai, Japan), with three cycles of 60 s ON (2,500 rpm) and 10 s OFF. For cell lysis, the ISOGEN reagent (Nippon Gene, Japan) was used. RT-qPCR was performed using 1 µg each of total RNA as template for reverse transcription with Super Script III (Thermo Fisher Scientific, USA). Then, 4 µL (from 100 µL cDNA sample) was used for RT-qPCR with the primer sets listed in Table S1. We used 16S rRNA as the reference gene. The Fast SYBR Green Master Mix (Thermo Fisher Scientific, USA) and the StepOnePlus Real-Time PCR System (Thermo Fisher Scientific, USA) were used for RT-qPCR. The $\Delta\Delta C_T$ method (81) was used to quantify the expression levels. It should be noted, in pilot experiments, no clear circadian accumulation rhythm in the expression of each gene was observed. Thus, it is presumed that each gene is expressed throughout the day at a transcription level corresponding to dusk (the light sample) in this study, even in cells under continuous light conditions that show similar colony patterns.

## Quantification of c-di-GMP

Each strain of *Leptolyngbya* was inoculated into square plates and cultured as was written for RNA extraction. Cells at hour 12 (the "light" sample) in the light after two LD cycles were detached from each culture plate after the addition of 10 mL DW. Cells were

centrifuged at 4°C, for 5 min, at 2,380 × $g$, supernatant was removed, and cells were washed three times. Then, cell pellets were resuspended in 10 mL of ice-cold DW, and the $OD_{730}$ value was measured; the required volume of suspension was collected in 2-mL screw tubes so that the value of the multiplication of the $OD_{730}$ value by the liquid volume (mL) was 3.0. The cells were pelleted by centrifugation at 4°C, 5 min, 20,000 × $g$ and stored at −80°C until use. c-di-GMP was extracted and purified as described by Kameda et al. (82) with some modifications. Bacterial suspensions were prepared to contain 25 nmol of cXMP as an internal control and adjusted to 100 µL. After incubation at 98°C for 10 min, the samples were washed twice by centrifugation at 4°C, 10 min, 18,000 × $g$ with 70% ethanol to collect the supernatant plus ethanol. After washing, the sample was evaporated to dryness and resuspended in 200 µL of water. This sample was washed with phenol and chloroform/isoamyl alcohol to yield approximately 200 µL of purified c-di-GMP sample. For analysis, 20 µL of the sample was injected into a liquid chromatography system (Waters 2690 Separations Module, Waters, Milford, USA). A Cosmosil 518-AR-II column (Nacalai tesque, Kyoto, Japan) was used as the column and a SecurityGuard Guard Cartridge Kit (Phenomenex, Torrance, CA) as the guard column. For all analyses, samples were analyzed using gradient mode, whereby the starting conditions were 97% water and 3% methanol and were changed to 25% water and 75% methanol at a flow rate of 0.5 mL/min for 25 min after which the gradient was returned to the starting conditions for a total run time of 40 min. The sample was transferred by electrospray to a triple quadrupole mass spectrometer (Quatro Ultima, Waters-Micromass, Manchester, UK) using multiple reaction monitoring mode to monitor the $m/z$ value transition for c-di-GMP, 691 > 152, and cXMP, 347 > 153. For the c-di-GMP spiking experiments, the WT cell extracts were used as matrix. Mass spectrometry analysis of 20 µL of this matrix with c-di-GMP in the range of 1–30 pmol confirmed a concentration-dependent increase in the c-di-GMP quantitative signal ($R$ = 0.9987). This calibration curve was then used to compare c-di-GMP in each strain.

## Database search

Sequences for nine putative DGC genes (*LBDG_00200, LBDG_02920, LBDG_10020, LBDG_31460, LBDG_34320, LBDG_34280, LBDG_44250, LBDG_28070*, and *LBDG_53130*) were obtained from the NCBI or Kyoto Encyclopedia of Genes and Genomes (KEGG) database. Simple Modular Architecture Research Tool (http://smart.embl-hei-delberg.de/smart/set_mode.cgi) was mainly used to search for domains. In addition, NCBI's Conserved Domains (https://www.ncbi.nlm.nih.gov/Structure/cdd/wrpsb.cgi) and EMBL-EBI's Pfam (https://pfam.xfam.org/) were employed. We performed a BLASTp search based on Dgc2 sequence in *Leptolyngbya*. Although we limited the bacterial phylum, we searched for genes encoding proteins in which CHASE2 and GGDEF domains and the transmembrane region are conserved, and compared the amino acid sequences of nine identified genes and *Leptolyngbya dgc2*. Multiple alignments were generated with the SnapGene software (from GSL Biotech; available at snapgene.com) with the Clustal Omega, MAFFT, MUSCLE, or T-coffee tool. We searched for orthologs of *N. punctiforme* motility-related genes in *Leptolyngbya* using BLAST Search (https://www.Genome.jp/tools/blast/) provided by KEGG. Genes with the best two-way hits and $e$ values < $1e^{-6}$ were considered orthologs and are summarized in Table 6.

## Resequencing of *Leptolyngbya* E22m1′ genome

The genomic DNA library was constructed using the tagmentation method (Nextera XT, Illumina, San Diego, CA, USA) and selected with an average insert size of 400–1,000 bp using AMPure XP beads. Sequencing of E22m1′ and E22m1-dg5 was performed with the MiSeq v3 600PE kit (Illumina). Sequencing data are available under accession numbers DRR277765 (E22m1-dg5) and DRR277766 (E22m1′), respectively. Breseq (version 0.36; 83) was used to find SNVs and indels, and E22m1-dg5 was used to eliminate background mutations in E22m1′. To find large indels and other genomic rearrangement

events, SV-Quest v1.0 (https://github.com/kazumaxneo/SV-Quest) was used with default settings.

## ACKNOWLEDGMENTS

We thank the members of the Iwasaki and Fujita laboratories for their valuable discussion. We also thank anonymous reviewers for addressing valuable questions and comments to improve the manuscript.

This work was supported by Grants-in-Aid from the Japanese Society for Promotion of Sciences (22520150, 25650111, 18K19349, 19K21608, 19H01221, and 20H01203 to H.I.).

The funding bodies had no role in the design of the study, collection, analysis, and interpretation of data or in writing the manuscript.

K.T. and H.I. conceived and designed the experiments; Y.F. isolated the original E22m1′ mutant; K.T., W.K., Ka.I., H.Y., K.Y., K.U., R.K., S.K., and Ku.I. performed the experiments; K.T., Y.F., S.K., and H.I. analyzed the data; and K.T. and H.I. wrote the manuscript. All authors read and approved the final manuscript.

## AUTHOR AFFILIATIONS

[1]Department of Electrical Engineering and Bioscience, Graduate School of Sciences and Engineering, TWIns, Waseda University, Tokyo, Japan
[2]Center for Gene Research, Nagoya University, Nagoya, Japan
[3]Graduate School of Nanobioscience, Yokohama City University, Yokohama, Japan
[4]Graduate School of Bioagricultural Sciences, Nagoya University, Nagoya, Japan
[5]metaPhorest, Bioaesthetics Platform, Waseda University, Tokyo, Japan

## AUTHOR ORCIDs

Kazuma Toida  http://orcid.org/0009-0001-9997-4408
Hiroki Yamamoto  http://orcid.org/0000-0002-3746-1260
Hideo Iwasaki  http://orcid.org/0000-0002-3754-4955

## FUNDING

| Funder | Grant(s) | Author(s) |
| --- | --- | --- |
| MEXT \| Japan Society for the Promotion of Science (JSPS) | 20H01203, 19H01221, 19K21608, 18K19349, 25650111, 22520150 | Hideo Iwasaki |

## AUTHOR CONTRIBUTIONS

Kazuma Toida, Conceptualization, Data curation, Investigation, Writing – original draft | Wakana Kushida, Investigation | Hiroki Yamamoto, Investigation, Writing – review and editing | Kyoka Yamamoto, Investigation | Kaichi Ishii, Investigation | Kazuma Uesaka, Investigation, Writing – review and editing | Robert A. Kanaly, Investigation, Writing – review and editing | Shinsuke Kutsuna, Data curation, Investigation, Writing – review and editing | Kunio Ihara, Investigation, Methodology, Writing – review and editing | Yuichi Fujita, Conceptualization, Investigation, Resources, Writing – review and editing | Hideo Iwasaki, Conceptualization, Funding acquisition, Investigation, Project administration, Resources, Supervision, Writing – original draft

## DATA AVAILABILITY

The strain used in the current study is available on reasonable request from the corresponding author.

## ADDITIONAL FILES

The following material is available online.

### Supplemental Material

**Supplemental file 1 (Spectrum04837-22-s0001.pdf).** Tables S1 to S3, Fig. S1 to S13, and and legends to movies.

**Movie S1 (Spectrum04837-22-s0002.mp4).** Time-lapse images of motility at the single filament level of the wild type (upper left), dgc2⁻ (upper middle), Δdgc2 (lower left), and Δdgc2;dgc2ex (lower right) strains on solid media for 20 min.

**Movie S2 (Spectrum04837-22-s0003.mp4).** Observation of the wild type (panel A) and Δdgc2 (panel B) strains for 100 h immediately after inoculation.

**Movie S3 (Spectrum04837-22-s0004.mp4).** Collective behavior of the dgc2⁻ (A) and Δdgc2 (B to D) strains on solid media.

**Movie S4 (Spectrum04837-22-s0005.mp4).** Behaviors of a wandering comet-like (A) and a rotating disk-like (B) clusters are shown (each extracted from Movie S3B).

**Movie S5 (Spectrum04837-22-s0006.mp4).** Transition of migrating cluster types. (A) Three comets unified into a larger one. (B) Integration of a comet into a disk. (C) Separation of a comet from a disk.

**Movie S6 (Spectrum04837-22-s0007.mp4).** Three types of disks showed three different behaviors. (A) A disk that was hit by comets multiple times but continued to rotate in the CCW direction for 300 min; (B) a disk that rotated in the CW direction, emitted comet-like clusters, and collapsed; and (C) a disk that rotated in the CCW direction, but changed direction to CW at 300-350 min, eventually disintegrating to release comet-like clusters.

**Movie S7 (Spectrum04837-22-s0008.mp4).** Rotation of a disk disturbed by collision with comets. The disk collided with other comets at ~185 and 400 min, and comets popped out at ~300 and 840 min.

**Movie S8 (Spectrum04837-22-s0009.mp4).** EPS released from gliding filaments of Δdgc2. India ink was dropped on the filaments of the Δdgc2 strain. The trajectory of the filament was stained black with the ink particles.

### Open Peer Review

**PEER REVIEW HISTORY (review-history.pdf).** An accounting of the reviewer comments and feedback.

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
