## [Reviewer comments · Microbiology Spectrum]

Microbiology Spectrum

The GGDEF protein Dgc2 suppresses both motility and biofilm formation in the filamentous cyanobacterium *Leptolyngbya boryana*.

Kazuma Toida, Wakana Kushida, Hiroki Yamamoto, Kyoka Yamamoto, Kaichi Ishii, Kazuma Uesaka, Robert Kanaly, Shinsuke Kutsuna, Kunio Ihara, Yuichi Fujita, and Hideo Iwasaki

Corresponding Author(s): Hideo Iwasaki, Waseda Daigaku

Review Timeline:

Submission Date:	December 31, 2022
Editorial Decision:	February 8, 2023
Revision Received:	June 17, 2023
Accepted:	June 30, 2023

Editor: Erik Hom

Reviewer(s): Disclosure of reviewer identity is with reference to reviewer comments included in decision letter(s). The following individuals involved in review of your submission have agreed to reveal their identity: Thomas Wallner (Reviewer #2)

Transaction Report:

DOI: <https://doi.org/10.1128/spectrum.04837-22>

February 8, 2023

Prof. Hideo Iwasaki
Waseda Daigaku
Dept. Electrical Engineering & Biosciences
TWIns 1F, 2-2 Wakamatsu, Shinjuku
Tokyo
Japan

Re: Spectrum04837-22 (The *dgc2* gene encoding di-guanylate cyclase suppresses both motility and biofilm formation in the filamentous cyanobacterium *Leptolyngbya boryana*.)

Dear Prof. Hideo Iwasaki:

Thank you for submitting your manuscript to Microbiology Spectrum. Your manuscript has now been read by two expert reviewers and their comments are attached. Both found your manuscript interesting, but have constructive feedback for improvement and revision. In particular, both reviewers believe you need to be more conservative in your conclusions about the role of c-di-GMP given your data: please do not overstate what your results imply. At Microbiology Spectrum, we value results presented as evenly/fairly as possible and do not emphasize subjective assessments of "novelty" or "importance" as a criterion for publication, so please do not feel the pressure to exaggerate or speculate beyond what your results plainly imply.

When submitting a revised version of your paper, please provide (1) in a cover letter to me, a brief summary of what you have changed in your manuscript, (2) in a separate file, point-by-point responses to the issues raised by the reviewers as file type "Response to Reviewers," not in your cover letter, and (3) a PDF file that indicates the changes from the original submission (by highlighting or underlining the changes) as file type "Marked Up Manuscript - For Review Only". Please use this link to submit your revised manuscript - we strongly recommend that you submit your paper within the next 60 days or reach out to me. Detailed instructions on submitting your revised paper are below.

Link Not Available

Sincerely,

Erik Hom

Journals Department
Reviewer comments:

Reviewer #1 (Comments for the Author):

In this work, the authors reported that a mutation in the *dgc2* gene renders the ability of motility and biofilm formation in the filamentous cyanobacterium *Leptolyngbya boryana*. The rigorous genetic and microscopic characterizations support the idea that the c-di-GMP-producing DGC2 protein inhibits motility and biofilm formation in their wild-type strain. The manuscript is well-written, the explanations and logic are clear, and many interesting and quantitative data well support the conclusions. The authors could consider the following minor points to be addressed to improve the manuscript.

Line 53: Ref (20) could be wrong.

Line 128: "show exhibit" remove either verb.

Line 170: The authors could briefly mention whether the cellular movement is phototactic or not, as many cyanobacteria exhibit phototactic motility.

Line 243: Though the mechanism of preference for CCW in *Leptolyngbya* movement is reasonably out of the scope of the current manuscript, the authors could explicitly mention that they did the experiments in the northern hemisphere.

Line 285: "we initially confirmed that *dgc2* expression is exclusively observed in the $\Delta dgc2$ strain under solid media." What did the author mean? Why is it transcribed when the gene is gone?

Line 295: Given the result of Figure 7C, the authors could consider weakening the title of this chapter, as the GGDEF domain is not the primarily responsible region for complementing the phenotypes.

Line 306: "Compared with the" check the font size

Line 314: "GGDE" -> "GGDEF"

Line 342: "This was surprising..." The authors did not observe a decrease in c-di-GMP levels in the $\Delta dgc2$ strain (Figure 8).

Therefore, the authors could be more cautious with such a statement. For example, the disruption of the *dgc2* gene could activate type IV pili in a c-di-GMP-independent manner, and then the active surface motility with type IV pili led to a stable establishment of a biofilm-like structure on a glass surface, as adhesive pili are known to be crucial for biofilm formation of *Pseudomonas* or *Clostridium*. Indeed Figure 7C ($\Delta dgc2:dgc2[\Delta cat]ex$) apparently implies that the GGDEF domain of Dgc2 is not crucial for complementing the disruption of *dgc2*.

Line 368: The authors used the *dg5* strain as "wild-type" in this work, but it is already one of the spontaneous "mutant" acquiring the ability to grow in the dark in the presence of carbon sources. The authors could comment how the current findings could be relevant in an ecological setting. Does the same mutation render the same motility phenotype in the original "wild-type" background?

Line 400: "which contrasted..." The same as line 342.

Line 402: Add a reference.

Line 404: "T4P, which is required for motility, may also promote adhesion of cells to solid-phase surfaces." This is already known in other bacteria. Please add some references.

The references should be updated. E.g., (7), (9), (10) are gone. (56): 2002->2020.

Figure 1D: Why did the authors detect the transcript of *dgc2* from Δ the *dgc2* strain, as almost all the ORF region is deleted (Figure 1B)?

Figure 2B: Verocity. Explain more details about the calculation. Total distances of the movement? Or displacement of the cells at the final frame from the first frame? What is the frame rate of the acquired movies, as it is crucial regarding the significant digits? Moreover, from Movie S1, some cells of *dgc-* strain seem immotile. Is that true also for $\Delta dgc2$ -? Please add a short comment on that.

Figure 7A: What is the small orange rectangle?

Movie S2: $\Delta dgc2$ strain moved better after 1 d incubation after spotting (as this time point is used by the authors for e.g., Fig 2B). The cell movement looked worse at 2 d, but again got better at 3 d (or more). Could the authors possibly comment on possible reasons for that?

Reviewer #2 (Comments for the Author):

The manuscript by Toida et al., describes the isolation of a spontaneous gain of function mutation in *Leptolyngbya boryana*, a filamentous cyanobacterium. The commonly used lab strain of *Leptolyngbya boryana* is not motile, where the authors isolated a mutant (*dgc2-*) that shows motility of the filaments and unique colony morphologies. The authors used whole genome sequencing to identify potential mutations that led to the observed phenotype. They could identify a transposon insertion into a putative diguanylate cyclase gene (*dgc2*). The responsibility of the *dgc2* gene for the observed phenotypes was confirmed using an independently generated $\Delta dgc2$ mutant and various complementation assays which included overexpression of the *dgc2* gene as well as catalytically inactive variants.

The second messenger c-di-GMP, produced by diguanylate cyclases, is known to be involved in the lifestyle decisions (motility, biofilm formation, biofilm dispersion and virulence) in many bacteria. Low cellular levels of c-di-GMP are known to facilitate motility, whereas a high c-di-GMP inhibits motility and promotes biofilm formation. Interestingly, the authors describe that the *dgc2* mutant of *Leptolyngbya boryana* shows enhanced surface attachment and has decreased content of c-di-GMP compared

to the wild type lab strain. Therefore, they conclude a novel role of c-di-GMP in biofilm formation in *Leptolyngbya boryana*. The authors intensively analyze the motility of $\Delta dgc2$ filaments and their unique colony patterns in a quantitative and qualitative manner using time-lapse imaging, kymograph analyses and tracking of trajectories combined with robust statistical analyses. They further show that the mutant strain secretes a significant higher amount of EPS than the wild type lab strain. Toida et al. further compared the transcript abundance of genes that are assigned with type IV pili, assembly of these cell appendages and secretion of polysaccharides in filamentous cyanobacteria. The abundances of the pilA mRNA, major subunit of type IV pili, and the hspE mRNA showed a light dependent increase in the $\Delta dgc2$ mutant.

Major points:

The authors conclude based on their experimental results that the Dgc2 protein plays a role in suppressing both gliding motility and biofilm formation. However, the presented results are partially lacking mechanistic details of the c-di-GMP dependent regulation and biofilm formation in *Leptolyngbya boryana*. How does the only slightly lowered c-di-GMP content in the mutant strain affect the biofilm formation and enhance surface attachment while simultaneously enhancing motility on agar plates? Overexpression of an EAL-domain containing c-di-GMP phosphodiesterase (like YhjH from *E. coli*) in *Leptolyngbya* could help to demonstrate the novel role of c-di-GMP in this cyanobacterium. The authors provide different scenarios to explain the observed phenotype of the $\Delta dgc2$ mutant. The use of fluorescently labelled protein involved in gliding motility and polysaccharide secretion may clarify whether the Dgc2 protein interacts directly with the motility apparatus or acts in more globally.

The authors provide evidence for an enhanced secretion of polysaccharides based on their transcriptome analyses and EPS staining; but the effect of the ~5-fold increase in the pilA mRNA abundance is not considered in detail. Are there known c-di-GMP-related domains coupled to transcription factors or other components of signaling pathways?

Also, different culture condition (constant light versus diurnal light-dark cycles) for the time-lapse imaging and the transcriptome analyses/ c-di-GMP quantification were used, making the interpretation of the presented results more difficult. Diurnal changes can have an influence on the overall RNA amount of the cell.

The results of the quantification of c-di-GMP in WT and $\Delta dgc2$ strain seem to be close to lower limit of quantification and only two replicates (biological/technical?) were used. The authors might try to optimize the method to achieve more reliable results.

The massive overexpression of the catalytically inactive GGDEF domain of Dgc2 could sequester the cellular c-di-GMP as the i-site of the GGDEF domain is still present. The use of the native promoter only might give more clear results. Also, watching the movie S3 and the figure S6, I had the impression that there is a difference in the occurrence/frequency of comets and disks in the $dgc2^-$ strain compared to the $\Delta dgc2$ strain. Did the authors check whether the *dgc2* gene is still transcribed in the $dgc2$ mutant?

The authors may provide more information on the physiological role of the observed colony patterns formation and their dynamic. In addition, when watching movie S2, I noticed some kind of temporal periodicity of the filament movement (phases of movement and phases of non-movement). Are the cells synchronized in regard to cell division?

Furthermore, a graphic summary of the findings and discussion would enhance the manuscript.

In principle, all references should be thoroughly checked again, since citations are present in the text, but not in actual reference library. In addition, some references refer to incorrect publication or publications that do not cover the specified topic. Here are just a few examples.

Line: 44: there are no reference #7, #9 and #10 in the reference library, but cited in the text

Line 53: reference #20 does not deal with type IV pili-based motility of *Synechocystis* at all, but with photosynthesis related genes

Line 285: "*dgc2* expression is exclusively observed in the $\Delta dgc2$ strain grown on solid media". I assume the authors are actually referring to the WT strain here?

Line 401: reference #64 specifically deals with the occurrence of c-di-GMP modulating output domains in cyanobacteria; *Pseudomonas aeruginosa* is not mentioned at all, only *Pseudomonas stutzeri* is mentioned once

Minor points:

All references include the web links to the corresponding websites. Is this data intended in this way?

Line 16: formation of bacteria is manifested in...

Line 24: encodes a protein containing the GGDEF motif, which is conserved in the catalytic domain...

Line 26: for this DGC

Line 34: lacking the DGC gene *dgc2* elicits...

Line 45: the basis for the formation of complex...

Line 109: without changing the amino acid sequence of the protein

Line 114: Thereby designating LBDG_02920 as *dgc2*... Shouldn't the assignment of gene names be based on the nomenclature for bacterial gene names? (*dgcA* to *dgcI* instead of *dgc1* to *dgc9*)

Line 118: No significant difference in cell morphology were observed at higher magnification.

Line 153: genetically different from the...

Line 273: encodes a subunit of a glycosyl transferase...

Line 277: except picocyanobacterial. The HmpF proteins forms a fibrous structure within...

Line 320: Since the UEA-I was also weakly observed...

Line 33: indicating that motility

Line 380: The dramatic increase

Line 382: Considering the presence of multiple copies of *dgc* genes in *Leptolyngbya*, the effect of a single deletion of the *dgc2* gene...

Line 384: While the mechanism is currently unknown, several scenarios...

Staff Comments:

Preparing Revision Guidelines

Please return the manuscript within 60 days; if you cannot complete the modification within this time period, please contact me. If you do not wish to modify the manuscript and prefer to submit it to another journal, please notify me of your decision immediately so that the manuscript may be formally withdrawn from consideration by Microbiology Spectrum.

The manuscript by Toida *et al.*, describes the isolation of a spontaneous gain of function mutation in *Leptolyngbya boryana*, a filamentous cyanobacterium. The commonly used lab strain of *Leptolyngbya boryana* is not motile, where the authors isolated a mutant (*dgc2*-) that shows motility of the filaments and unique colony morphologies. The authors used whole genome sequencing to identify potential mutations that led to the observed phenotype. They could identify a transposon insertion into a putative diguanylate cyclase gene (*dgc2*). The responsibility of the *dgc2* gene for the observed phenotypes was confirmed using an independently generated $\Delta dgc2$ mutant and various complementation assays which included overexpression of the *dgc2* gene as well as catalytically inactive variants.

The second messenger c-di-GMP, produced by diguanylate cyclases, is known to be involved in the lifestyle decisions (motility, biofilm formation, biofilm dispersion and virulence) in many bacteria. Low cellular levels of c-di-GMP are known to facilitate motility, whereas a high c-di-GMP inhibits motility and promotes biofilm formation. Interestingly, the authors describe that the *dgc2* mutant of *Leptolyngbya boryana* shows enhanced surface attachment and has decreased content of c-di-GMP compared to the wild type lab strain. Therefore, they conclude a novel role of c-di-GMP in biofilm formation in *Leptolyngbya boryana*.

The authors intensively analyze the motility of $\Delta dgc2$ filaments and their unique colony patterns in a quantitative and qualitative manner using time-lapse imaging, kymograph analyses and tracking of trajectories combined with robust statistical analyses. They further show that the mutant strain secretes a significant higher amount of EPS than the wild type lab strain.

Toida *et al.* further compared the transcript abundance of genes that are assigned with type IV pili, assembly of these cell appendages and secretion of polysaccharides in filamentous cyanobacteria. The abundances of the *pilA* mRNA, major subunit of type IV pili, and the *hspE* mRNA showed a light dependent increase in the $\Delta dgc2$ mutant.

Major points:

The authors conclude based on their experimental results that the Dgc2 protein plays a role in suppressing both gliding motility and biofilm formation. However, the presented results are partially lacking mechanistic details of the c-di-GMP dependent regulation and biofilm formation in *Leptolyngbya boryana*. How does the only slightly lowered c-di-GMP content in the mutant strain affect the biofilm formation and enhance surface attachment while simultaneously enhancing motility on agar plates? Overexpression of an EAL-domain containing c-di-GMP phosphodiesterase (like YhjH from *E. coli*) in *Leptolyngbya* could help to demonstrate the novel role of c-di-GMP in this cyanobacterium. The authors provide different scenarios to explain the observed phenotype of the $\Delta dgc2$ mutant. The use of fluorescently labelled protein involved in gliding motility and polysaccharide secretion may clarify whether the Dgc2 protein interacts directly with the motility apparatus or acts in more globally.

The authors provide evidence for an enhanced secretion of polysaccharides based on their transcriptome analyses and EPS staining; but the effect of the ~5-fold increase in the *pilA* mRNA abundance is not considered in detail. Are there known c-di-GMP-related domains coupled to transcription factors or other components of signaling pathways?

Also, different culture condition (constant light versus diurnal light-dark cycles) for the time-lapse imaging and the transcriptome analyses/ c-di-GMP quantification were used, making the

interpretation of the presented results more difficult. Diurnal changes can have an influence on the overall RNA amount of the cell.

The results of the quantification of c-di-GMP in WT and $\Delta dgc2$ strain seem to be close to the lower limit of quantification and only two replicates (biological/technical?) were used. The authors might try to optimize the method to achieve more reliable results.

The massive overexpression of the catalytically inactive GGDEF domain of Dgc2 could sequester the cellular c-di-GMP as the i-site of the GGDEF domain is still present. The use of the native promoter only might give more clear results. Also, watching the movie S3 and the figure S6, I had the impression that there is a difference in the occurrence/frequency of comets and disks in the *dgc2*-strain compared to the $\Delta dgc2$ strain. Did the authors check whether the *dgc2* gene is still transcribed in the *dgc2* mutant?

The authors may provide more information on the physiological role of the observed colony patterns formation and their dynamic. In addition, when watching movie S2, I noticed some kind of temporal periodicity of the filament movement (phases of movement and phases of non-movement). Are the cells synchronized in regard to cell division?

Furthermore, a graphic summary of the findings and discussion would enhance the manuscript.

In principle, all references should be thoroughly checked again, since citations are present in the text, but not in the actual reference library. In addition, some references refer to incorrect publications or publications that do not cover the specified topic. Here are just a few examples.

Line: 44: there are no references #7, #9 and #10 in the reference library, but cited in the text

Line 53: reference #20 does not deal with type IV pili-based motility of *Synechocystis* at all, but with photosynthesis related genes

Line 285: "*dgc2* expression is exclusively observed in the $\Delta dgc2$ strain grown on solid media". I assume the authors are actually referring to the WT strain here?

Line 401: reference #64 specifically deals with the occurrence of c-di-GMP modulating output domains in cyanobacteria; *Pseudomonas aeruginosa* is not mentioned at all, only *Pseudomonas stutzeri* is mentioned once

Minor points:

All references include the web links to the corresponding websites. Is this data intended in this way?

Line 16: formation of bacteria is manifested in...

Line 24: encodes a protein containing the GGDEF motif, which is conserved in the catalytic domain...

Line 26: for this DGC

Line 34: lacking the DGC gene *dgc2* elicits...

Line 45: the basis for the formation of complex...

Line 109: without changing the amino acid sequence of the protein

Line 114: Thereby designating *LBDG_02920* as *dgc2*... Shouldn't the assignment of gene names be based on the nomenclature for bacterial gene names? (*dgcA* to *dgcI* instead of *dgc1* to *dgc9*)

Line 118: No significant difference in cell morphology were observed at higher magnification.

Line 153: genetically different from the...

Line 273: encodes a subunit of a glycosyl transferase...

Line 277: except picocyanobacterial. The HmpF proteins forms a fibrous structure within...

Line 320: Since the UEA-I was also weakly observed...

Line 33: indicating that motility

Line 380: The dramatic increase

Line 382: Considering the presence of multiple copies of *dgc* genes in *Leptolyngbya*, the effect of a single deletion of the *dgc2* gene...

Line 384: While the mechanism is currently unknown, several scenarios...

June 30, 2023

Prof. Hideo Iwasaki
Waseda Daigaku
Dept. Electrical Engineering & Biosciences
TWIns 1F, 2-2 Wakamatsu, Shinjuku
Tokyo
Japan

Re: Spectrum04837-22R1 (The GGDEF protein Dgc2 suppresses both motility and biofilm formation in the filamentous cyanobacterium *Leptolyngbya boryana*.)

Dear Prof. Hideo Iwasaki:

Your manuscript has been accepted, and I am forwarding it to the ASM Journals Department for publication. You will be notified when your proofs are ready to be viewed.

I would want you to add a bit more text/clarification in your final version of proofs concerning your response to Review#2: "Certainly, in terms of the naming convention, it would be more appropriate to use the alphabet. However, in cyanobacteria, some PAS-containing GGDEF genes has been already named dgcA-dgcD in *Synechocystis*, which do not necessarily correspond to out list, and if we change the name to dgcA-dgcI as it is now, new confusion will arise. Although there is a discrepancy with the strict naming convention, we would like to adopt the practice of numbering the genes by numbers for convenience. If the editorial office decides that we must use the alphabet, we will reconsider."

Can you please make sure to include some text in your final manuscript that describes this nomenclature discrepancy? I am not concerned by your choice of naming convention but want to make sure that your contribution will be most useful by making sure to reference gene name synonyms where appropriate that would be helpful in bridging your findings with those in the community who might use a different naming convention. This is a minor edit, which I feel should not delay acceptance and is one I think you would understand how to implement without further editorial review.

Sincerely,

Erik Hom
Editor, Microbiology Spectrum
